# Temporal stratification of amyotrophic lateral sclerosis patients using disease progression patterns

Daniela M. Amaral [1,2], Diogo F. Soares [1] ✉, Marta Gromicho [3], Mamede de Carvalho[3], Sara C. Madeira [1], Pedro Tomás [4] & Helena Aidos [1] ✉

Identifying groups of patients with similar disease progression patterns is key to understand disease heterogeneity, guide clinical decisions and improve patient care. In this paper, we propose a data-driven temporal stratification approach, ClusTric, combining triclustering and hierarchical clustering. The proposed approach enables the discovery of complex disease progression patterns not found by univariate temporal analyses. As a case study, we use Amyotrophic Lateral Sclerosis (ALS), a neurodegenerative disease with a non-linear and heterogeneous disease progression. In this context, we applied ClusTric to stratify a hospital-based population (Lisbon ALS Clinic dataset) and validate it in a clinical trial population. The results unravelled four clinically relevant disease progression groups: slow progressors, moderate bulbar and spinal progressors, and fast progressors. We compared ClusTric with a state-of-the-art method, showing its effectiveness in capturing the heterogeneity of ALS disease progression in a lower number of clinically relevant progression groups.

Amyotrophic Lateral Sclerosis (ALS) is a neurodegenerative disease consisting of the progressive loss of motor neurons, which results in progressive weakness and, eventually, in a patient's death due to respiratory failure[1]. There is no cure, the current treatments aim to prolong survival and quality of life. The life expectancy of a patient diagnosed with ALS is on average 3–5 years[2]. Furthermore, ALS has high variability in the progression rate and clinical phenotype[3], being a challenge to determine the optimal time for medical interventions (such as non-invasive ventilation) and to ensure if a specific treatment is effectively slowing disease progression.

After diagnosis, patients with ALS are regularly followed up, using tools for assessing their functional disability, respiratory function, and relevant clinical events. The Revised ALS Functional Rating Scale (ALSFRS-R)[4] is a questionnaire-based scale that measures and tracks changes in the patient's functional ability over time. This score can be divided into subscores related to the bulbar, limb, trunk, and respiratory domains. The analysis of the collected longitudinal data can lead to the extraction of temporal patterns, which can help clinicians to better understand disease progression.

However, the analysis of these subscores to measure patients' progression is not trivial. Patients are frequently evaluated at different stages of their disease, and although there is a general decline in their condition over time, there may be periods of mild improvement and stability[5]. The identification of common disease progression patterns can improve patient stratification, which deals with the disease's heterogeneity. These groups can be useful to define clinical trial criteria and improve prognostic predictions to aid clinical decisions and supportive care[6].

Several ALS stratification studies split patients according to some clinical features or use traditional unsupervised learning techniques that do not take full advantage of the temporal dimension of data in the patient's records[7,8]. These works often assume that the progression

[1]LASIGE, Faculdade de Ciências, Universidade de Lisboa, Lisboa, Portugal. [2]Instituto Superior Técnico, Universidade de Lisboa, Lisboa, Portugal. [3]Instituto de Medicina Molecular e Faculdade de Medicina, Universidade de Lisboa, Lisboa, Portugal. [4]INESC-ID, Instituto Superior Técnico, Universidade de Lisboa, Lisboa, Portugal. ✉e-mail: dfsoares@ciencias.ulisboa.pt; haidos@ciencias.ulisboa.pt

in ALS is linear, despite the evidence that it can be non-linear and vary across disease severity[9].

The progression rate is widely used to stratify ALS patients (fast, moderate, and slow). It is calculated between the maximum ALSFRS-R score (48) and the ALSFRS-R score at diagnosis, divided by disease duration (in months). The distribution of values in the progression rate can be used to group patients by defining cut-offs[10,11].

Clinical stages are also potential ways to stratify patients. These clinical stages are scoring systems used to allocate patients into specific disease phases based on their features. There are three main staging systems in ALS: Kings'[12], MiToS[13], and FT9[14]. Although often used to stratify patients[15,16], these scoring systems group patients using a set of diagnosis or functional features, without considering patients' follow-up.

Clustering algorithms are usually used to tackle stratification problems. Pires et al.[6] used the Expectation-Maximization algorithm (EM) to cluster ALS patients, creating Patient profiles. These patient profiles were then used to learn specialized prognostic models, outperforming other approaches without stratification for several prediction windows. Since both the stratification and the predictive models use only one appointment, they disregard the temporal dynamics of disease progression. The same applies to ensemble approaches[17].

Several non-linear parametric models[18–20] were used to study the ALS disease progression. However, since these approaches are restricted to model predefined trajectory shapes, they may fail to identify some disease progression patterns. Ramamoorthy et al.[9] characterized disease progression in ALS using an approach based on a mixture of Gaussian processes (MoGP). Their method uses Gaussian process regression to allow the identification of non-linear trajectory progression patterns, a Dirichlet process mixture model to identify clusters in data, and a Monotonic Inductive Bias to encourage declining trajectories. The model was able to learn clusters of patients sharing similar disease progression patterns. The authors concluded that the patients could have linear and non-linear trajectories, motivating the use of non-parametric models that could capture both trajectories.

In recent years, several pattern-based approaches were used to study ALS disease progression. Matos et al.[21] applied a biclustering-based approach to uncover disease presentation patterns. Martins et al.[22] proposed to use itemset mining with sequential pattern mining to unravel disease presentation and progression patterns using static and temporal data. Soares et al.[23] proposed BicTric, a classifier able to learn predictive models from static and temporal data using discriminative patterns[24] obtained using biclustering and triclustering[25]. Recently, Soares et al.[26] enhanced BicTric with `TCtriCluster`, a triclustering algorithm incorporating temporal contiguity constraints. All these approaches used temporal preprocessing using snapshots and the time windows approach proposed by Carreiro et al.[27] and were used to learn predictive models for different clinically relevant ALS endpoints.

In this work, we propose ClusTric, a data-driven methodology that stratifies patients based on temporal data and without any linearity assumption. Our method combines triclustering and hierarchical clustering and is three-fold: (1) learn disease progression patterns using a triclustering algorithm, (2) compute the representative patterns of the obtained triclusters, and (3) find the progression groups using hierarchical clustering. We applied the proposed methodology to stratify a hospital-based dataset of ALS patients and validated it in a clinical trial ALS dataset. We characterized the identified disease progression groups by assessing the trajectories of key features and performed a survival analysis considering an 8-year follow-up. Additionally, we studied how the patients' disease progression evolved in a 6-month follow-up. Our results show the effectiveness of our approach and improvements when compared to a state-of-the-art model based on the aggregation of patient trajectories[9].

## Results

We performed experiments with our proposed method, ClusTric, in the Lisbon ALS Clinic Dataset and verified the outcomes through external validation using the Pooled Resource Open-Access ALS Clinical Trials (PRO-ACT) cohort[28]. We conducted an extensive analysis of ClusTric results regarding patients' follow-up and studied how the patients' progression evolved in 6 months of follow-up by studying each progression group. Moreover, we performed experiments using a state-of-the-art framework named MoGP, which consists of an unsupervised approach to aggregate patient trajectories into clusters using Gaussian processes, and determine the number of clusters using a Dirichlet process mixture model[9]. Finally, we explored the survival of the progression groups identified by both methodologies, ClusTric and MoGP. In this section, we present the results obtained.

### ALS datasets

We conducted our study using the Lisbon ALS Clinic dataset, which consists of Electronic Health Records from ALS Patients who have been regularly monitored at the local ALS clinic since 1995. The used dataset was last updated in May 2023 and includes 1677 patients. Each patient in the dataset has a set of static features (Table 1) such as

**Table 1 | Characterization of the population used in the case study**

| | PRO-ACT $n = 3880$ | Lisbon ALS $n = 983$ | $p$-value |
|---|---|---|---|
| Sex | | | $4.0 \times 10^{-6}$ |
| Male | 2460, 63.4% | 545, 55.4% | – |
| Female | 1420, 36.6% | 438, 44.6% | – |
| Age at onset (years) | | | 0 |
| Median, IQR | 57, 49–64 | 62, 54–71 | – |
| Average, Std | 56.20, 11.33 | 61.44, 12.39 | – |
| Diagnostic delay (months) | | | $2.8 \times 10^{-9}$ |
| Median, IQR | 9.05, 5.84–13.98 | 10.71, 6.04–17.58 | – |
| Average, Std | 11.30, 8.85 | 15.85, 20.19 | – |
| BMI at diagnosis (kg/m²) | | | 0 |
| Median, IQR | 26.22, 23.33–29.35 | 24.61, 22.60–27.23 | – |
| Average, Std | 26.63, 5.31 | 25.01, 13.98 | – |
| Onset form | | | $5.5 \times 10^{-11}$ |
| Spinal | 1919, 49.5% | 652, 66.3% | – |
| Bulbar | 562, 14.5% | 257, 26.1% | – |
| Generalized | 11, 0.3% | 13, 1.3% | – |
| Other | 355, 9.1% | 61, 6.2% | – |
| C9orf72 | | | |
| Yes | – | 41, 4.2% | – |
| No | – | 454, 46.2% | – |
| UMNvsLMN | | | |
| UMN | – | 166, 16.9% | – |
| LMN | – | 404, 41.1% | – |
| Both | – | 16, 1.6% | – |

Characterization of the population in the PRO-ACT dataset and the Lisbon ALS Clinic dataset. Numerical features are described using Median, Interquartile Range (IQR), Average, and Standard deviation (Std); categorical features are described using the number of patients, and percentage. Mann-Whitney $U$ two-sided test and $\chi^2$ two-sided test were used to assess the similarity of the distributions of the continuous and categorical variables, respectively, between the two datasets. Note that some features have missing values and some features are not present in PRO-ACT.

demographics, disease severity, co-morbidities, medication, genetic information, exercise and smoking habits, past trauma/surgery, occupations, and familial history. Sex was considered in the study design as it is an important attribute to be considered in the disease study. In the context of the medical evaluation, the sex of human research participants was determined by self-report as well as physical and physiological characteristics, such as genetics, hormone function, and reproductive anatomy. Additionally, there are temporal features collected repeatedly during follow-up, including disease progression tests such as the ALSFRS-R scale and respiratory tests. Patients are assessed with an average frequency of 3 months. The study was conducted in accordance with the Declaration of Helsinki and was approved by the local (Faculty of Medicine, University of Lisbon) ethics committee. Informed consent to participate in the study was obtained from all patients. No individual-level data was shared. No compensation was given for participation.

Additionally, we performed an external validation of the proposed stratification method on a publicly available repository of merged ALS clinical trials data, the Pooled Resource Open-Access ALS Clinical Trials dataset (PRO-ACT)[28]. PRO-ACT includes information from a specific population of 11675 ALS patients who participated in clinical trials. It provides a comprehensive range of data features, including demographic details, laboratory results, past medical and familial history, and more (Table 1). However, since multiple trials were merged to create this dataset, different types of information are available for different patients. The PRO-ACT data went through a preprocessing phase to calculate the ALSFRS-R subscores (Table 2) used as input for our methodology. The records were sorted based on the difference between the first time a patient was observed and the time of each assessment over the trial. The records without the ALSFRS-R score or its respiratory items were removed.

The physical condition of ALS patients hampers them from completing all the prescribed tests in a single day. Therefore, the dates of the different performed tests are misaligned. To tackle this problem, the Lisbon ALS Clinic Dataset went through a preprocessing phase, following the methodology proposed by Carreiro et al.[27].

This methodology considers the temporal distribution of tests by creating snapshots of the patient's condition. These snapshots group together tests that were performed within a clinically accepted time window, which in this case is set to 100 days (as in ref. 27), given the mean frequency of patient assessments.

To generate these snapshots, the process employed a hierarchical (agglomerative) clustering method with constraints, which is a state-of-the-art procedure for aligning data along a follow-up period[27]. The constraints applied during the grouping of evaluation sets followed the originally established principles, with just one constraint: all the tests within a snapshot must be different, as clinicians do not prescribe the same test in the same appointment. Other constraints previously considered relevant for predictive models were irrelevant here since we are dealing with an unsupervised method.

Finally, note that for the application of the proposed method, we have to understand our datasets as three-way data, with patients, features, and time as dimensions (as illustrated in Fig. 6). In particular, we considered the initial three appointments of the 7 features in Table 2 of the patients enrolled in the Lisbon ALS Clinic dataset and the PRO-ACT dataset. Patients with fewer than three appointments were excluded from the analysis, resulting in a total of 983 patients in the Lisbon ALS Clinic dataset and 3880 in the PRO-ACT dataset (see Tables 1 and S1 of the Supplementary material).

## ClusTric in Lisbon ALS clinic dataset

ClusTric was applied to the Lisbon ALS Clinic dataset. We performed experiments to determine the number of clusters, considering three, four and five clusters. The determination of the number of clusters was then guided by both the hierarchical clustering dendrogram (Fig. 1A), particularly by analyzing the inter-cluster difference, and by analyzing specific clustering metrics, namely the Silhouette score, and the Calinski-Harabsz and Davies-Bouldin indexes (Fig. 1B).

The resulting four patient clusters were characterized by analyzing the temporal patterns of key features, as defined by clinicians, over five appointments, covering approximately one year of follow-up (Fig. 1C). To ensure consistency in the trajectories, an onset-anchor value was added, representing the maximum clinical score assigned to the date corresponding to symptom onset, designated as time point 0. Based on the trajectories, we designated the clusters as Slow Progressors (SP), Moderate Progressors mainly bulbar (MPb), Moderate Progressors mainly spinal (MPs), and Fast Progressors (FP).

Furthermore, we examined the features at disease onset and at 1st visit of each cluster, and identified features that were more significant in distinguishing between the clusters compared to others (see Table 3). We applied the Chi-Square two-sided test to determine the difference between clusters in the proportion of sex. We also performed normality tests (Kolmogorov-Smirnov two-sided test) to determine which continuous features followed a normal distribution. For non-normal distribution features (age at onset, diagnostic delay, body mass index (BMI), the slope of ALSFRS-R at 6 months, ALSFRS-R at 1st visit, and phrenic nerve response latency at 1st visit) the non-parametric Kruskal–Wallis two-sided test was used to examine if the distribution of these features was the same for all the clusters. Pairwise comparisons between the clusters were performed with the Bonferroni correction. The vital capacity (VC) and the phrenic nerve response amplitude at 1st visit were the features following a normal distribution and the One-Way ANOVA was performed to compare the mean among the clusters in conjunction with a post-hoc Tukey's Honestly Significant Difference (HSD) test to identify specific pairwise differences between the clusters. The significance of each feature was determined using a $p$-value threshold of 0.05. For example, the BMI and the latency of the phrenic nerve at 1st visit were deemed non-significant in characterizing the patient clusters. Additionally, the significance of the remaining features varied among pairs of clusters. Specifically:

- The distribution of the diagnostic delay was only significantly different for the pair of clusters MPb and SP ($p$-value of 0.004).
- The age at disease onset exhibited the same distribution for clusters SP and MPs, while the distributions were

## Table 2 | ClusTric input features

| Feature | Description | Values range |
|---|---|---|
| ALSFRS-R | Functional score for monitoring disease progression | [0,48] |
| ALSFRSb | Functional subscore for monitoring bulbar progression | [0,12] |
| ALSFRSsUL | Functional subscore for monitoring upper limbs progression | [0,8] |
| ALSFRSsLL | Functional subscore for monitoring lower limbs progression | [0,8] |
| ALSFRSsT | Functional subscore for monitoring trunk progression | [0,8] |
| ALSFRSr | Functional subscore for monitoring respiratory progression | [0,12] |
| MITOS-stage | Score staging system based in ALSFRS-R | [0,4] |

Measurements of the 7 predefined temporal features considered, computed based on ALSFRS scores.

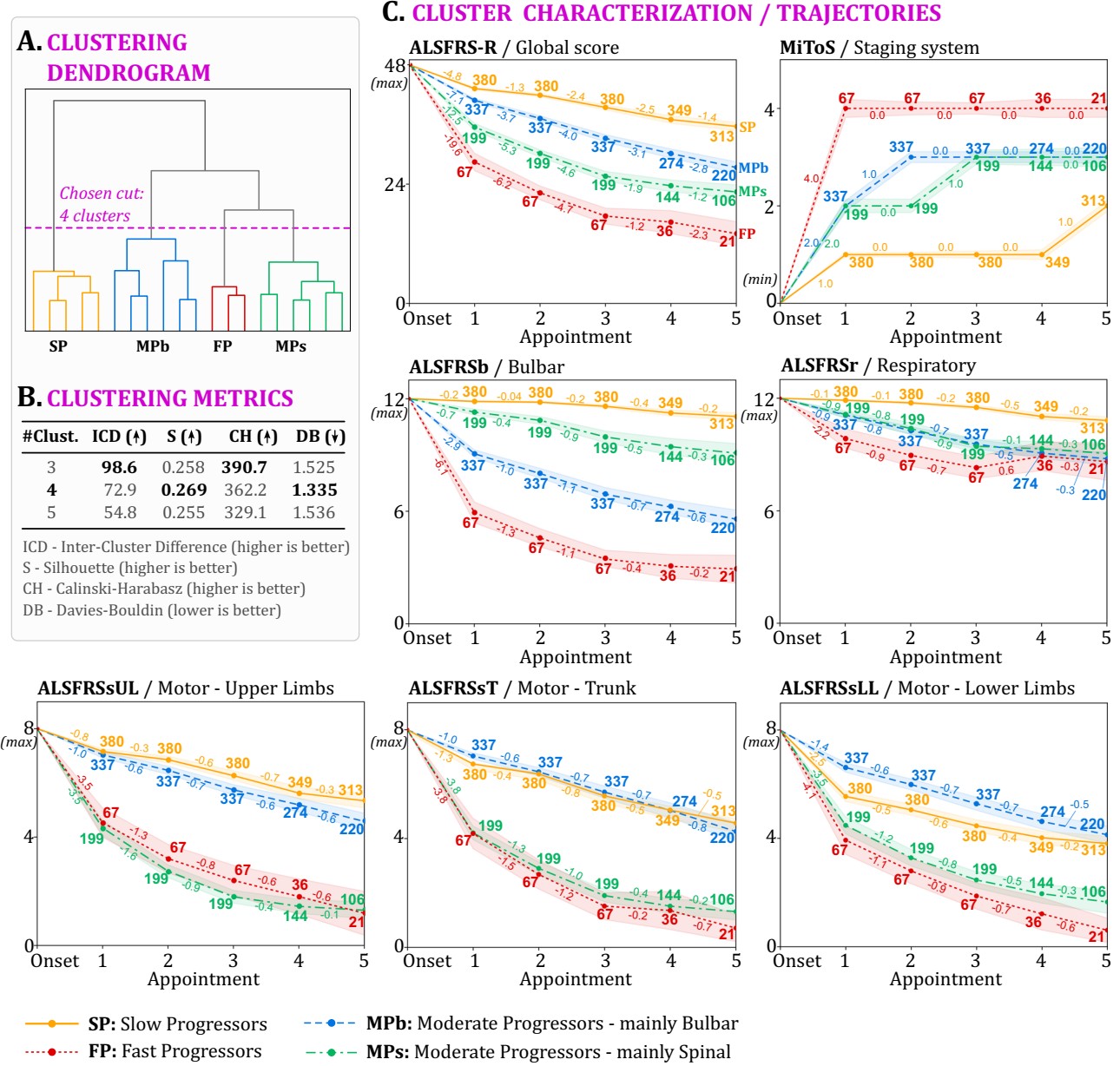

**Fig. 1 | Cluster analysis and characterization on Lisbon ALS cohort.**
**A** Dendrogram resulting from ClusTric; (**B**) clustering evaluation scores obtained with 3, 4, and 5 clusters; (**C**) average temporal feature trajectories (lines) and 95% confidence intervals (shades around lines). The numbers next to each point in the trajectories represent the number of patients in the < cluster,appointment > set, whereas the numbers between consecutive appointments indicate the average slope between consecutive measurements in a cluster. Source data are provided as a Source Data file.

significantly different for the other pairs of clusters. In particular, a *p*-value < 0.001 was found for all the remaining pairwise comparisons, except for the pair {MPb,FP} where it was 0.037.

- The difference in the vital capacity (VC) at 1st visit was not significant for the pair {MPb,MPs} (*p*-value of 0.692). In contrast, the VC for the cluster FP was notably lower than that of all the remaining clusters (*p*-value < 0.001 for the combinations involving cluster FP) and higher for the cluster SP (*p*-value < 0.001 for the combinations involving cluster SP).
- The slope of the ALSFRS-R score in the first 6 months showed significant differences only among the pairwise combinations involving cluster SP (*p*-value < 0.001 for the combinations involving cluster SP), similar to the mean phrenic amplitude at 1st visit.

- The ALSFRS-R at 1st visit distribution was significantly different in all the pairwise combinations.

The identified groups showed specific disease progressions that are representative of the patients' evolution, which cannot be understood as subtypes of the disease since a given patient could start progressing slower or faster at some point according to specific functional domains.

### ClusTric in PRO-ACT dataset
To validate the proposed methodology, ClusTric was also applied to the PRO-ACT dataset. As before, we learned the triclustering patterns, performed the triclustering-based data transformation and determined the number of clusters guided by the dendrogram resulting

**Table 3 | Disease onset and 1st visit characterization**

| Feature | Overall $N = 983$ | Cluster SP $N = 380$ | Cluster MPb $N = 337$ | Cluster MPs $N = 199$ | Cluster FP $N = 67$ | p-value |
|---|---|---|---|---|---|---|
| Male | 545 | 244 | 162 | 119 | 20 | $2.2 \times 10^{-8}$ |
|  | 55.4% | 64.2% | 48.1% | 59.8% | 29.9% |  |
| Age at onset (years) | 61.44 | 58.92 | 64.50 | 60.04 | 69.73 | $6.8 \times 10^{-13}$ |
|  | [60.62; 62.17] | [57.37; 60.47] | [62.99; 66.02] | [57.65; 62.42] | [66.52; 72.94] | SP = MPs |
| Diagnostic delay (months) | 15.85 | 17.35 | 13.17 | 16.96 | 17.53 | 0.006 |
|  | [14.59; 17.12] | [15.45; 19.25] | [11.46; 14.88] | [13.52; 20.40] | [10.34; 24.73] | SP ≠ MPb |
| BMI | 25.01 | 25.18 | 25.03 | 25.09 | 24.62 | 0.325 |
|  | [24.77; 25.26] | [24.77; 25.56] | [24.50; 25.56] | [24.32; 25.87] | [23.62; 25.62] |  |
| ALSFRS-R at 1st visit | 39.83 | 43.21 | 40.86 | 35.46 | 28.45 | 0 |
|  | [39.43; 40.17] | [42.92; 43.50] | [40.47; 41.24] | [34.71; 36.22] | [26.66; 30.23] | SP ≠ MPb ≠ MPs ≠ FP |
| Slope of ALSFRS-R at 6 months | −1.02 | −0.58 | −1.16 | −1.48 | −1.53 | 0 |
|  | [-1.09; -0.96] | [-0.65; -0.51] | [-1.26; -1.05] | [-1.66; -1.30] | [-1.82; -1.24] | SP ≠ {MPb, MPs, FP} |
| VC at 1st visit | 86.11 | 96.94 | 82.35 | 79.68 | 65.92 | $8.5 \times 10^{-24}$ |
|  | [84.29; 87.93] | [94.36; 99.51] | [79.60; 85.19] | [75.74; 83.61] | [59.16; 72.67] | MPb = MPs |
| Mean phrenic nerve amplitude at 1st visit | 0.52 | 0.67 | 0.43 | 0.50 | 0.42 | $3.3 \times 10^{-20}$ |
|  | [0.50; 0.54] | [0.63; 0.70] | [0.40; 0.46] | [0.45; 0.56] | [0.35; 0.49] | SP ≠ {MPb, MPs, FP} |
| Mean phrenic nerve latency at 1st visit | 8.32 | 8.21 | 8.43 | 8.31 | 8.24 | 0.111 |
|  | [8.21; 8.43] | [8.09; 8.34] | [8.27; 8.59] | [7.97; 8.66] | [7.66; 8.82] |  |

Characterization of each cluster (resulted from applying ClusTric to the Lisbon ALS Clinic dataset) according to selected features at disease onset and at 1st visit, and the significance level of each feature. Categorical features (sex) are represented by the number and percentage of patients within the cluster. Continuous features (age at onset, diagnostic delay, BMI, ALSFRS-R at 1st visit, Slope of ALSFRS-R at 6 months, VC at 1st visit, phrenic motor response amplitude and latency at 1st visit) are represented by mean value (first row for each variable) and 95% confidence interval (second row). The statistical tests used for data analysis were the Chi-square two-sided test for categorical features, the non-parametric Kruskal–Wallis two-sided test for non-normal distribution continuous features with the Bonferroni correction, and the One-Way ANOVA for normal distribution features with the Tukey's Honestly Significant Difference (HSD) test for pairwise comparisons.
*SP* slow progressors, *MPb* moderate progressors mainly bulbar, *MPs* moderate progressors mainly spinal, *FP* fast progressors.

from the application of hierarchical clustering (Fig. 2A) and of the different clustering metrics for three, four, and five clusters (Fig. 2B).

As performed for the Lisbon ALS Clinic dataset, the resulting four groups were characterized using the progression of the clinically relevant temporal features across five appointments (see Fig. 2C). The obtained trajectories show similarities with the ones observed in the Lisbon ALS Clinic dataset. However, the cluster MPs exhibits a slower progression in the Upper Limb subscore (ALSFRS-sUL) and a faster progression regarding the remaining motor subscores (ALSFRSsT and ALSFRSsLL). This difference arises because the measurements of the ALSFRSsUL subscore in the PRO-ACT dataset exhibit slight change over the considered timeframe when compared to the previously used Lisbon cohort (see Table S1 of the Supplementary material). Specifically, in the PRO-ACT dataset, the overall average ALSFRSsUL subscore is $5.80 \pm 1.92$ at the first appointment and $5.30 \pm 2.20$ at the third appointment. This contrasts with the Lisbon ALS Clinic dataset at equivalent time points, corresponding to $6.36 \pm 1.86$ at the first appointment and $4.92 \pm 2.59$ at the third appointment.

Another difference in the behavior of the cluster MPs when compared to the results from the Lisbon ALS Clinic dataset pertains to its MiToS stage's trajectory. In particular, in the Lisbon ALS Clinic dataset, the MiToS stage of the patients in this progression group remains 2 at the first and second appointments and increases to 3 at the third appointment (Fig. 1C). In contrast, in the PRO-ACT cohort (Fig. 2C), the MiTos stage of the cluster MPs remains constant at 1, over five consecutive appointments, mirroring the MiTos stage's trajectory of cluster SP until the third appointment.

Comparing the onset features of the disease progression groups identified in the two datasets (Lisbon ALS Clinic dataset and PRO-ACT), there are also slight differences (see Tables 3 and S2 of the Supplementary material). Particularly, in PRO-ACT, the cluster FP is predominantly composed of young man (61.9%) with a mean onset age of 54.87 years. Additionally, the patients comprising the cluster MPs have

a higher mean ALSFSR-R score at onset (39.08) compared to those in cluster MPb (35.60), which differs from the behavior verified in the Lisbon ALS Clinic dataset.

## Comparison with MoGP

MoGP[9] is a Gaussian process-based framework for aggregating patient trajectories into clusters with relevant results in ALS. This framework determines the number of clusters using a Dirichlet process mixture model and induces a monotonic bias to encourage the identification of declining trajectories. In this context, we compare ClusTric with MoGP. To ensure a fair analysis, we preprocessed the Lisbon ALS Clinic snapshots and the PRO-ACT following the same preprocessing steps imposed by MoGP[9]. This involved discarding patients who met any of the following criteria: (1) having fewer than three complete appointments, (2) exhibiting a difference of more than 7 years between the first visit and symptom onset, or (3) experiencing an increase of more than six points in ALSFRS-R between consecutive visits. Consequently, 954 patients from the Lisbon ALS Clinic dataset and 3801 patients from the PRO-ACT were included in this comparative experiment.

MoGP identified 27 clusters in the Lisbon ALS Clinic dataset, with the largest cluster comprising 75 patients and the smallest containing only 1 patient (Fig. 3A). To compare with ClusTric, we provide a visual representation of the four most representative cluster trajectories unveiled by MoGP (Fig. 3B). The cluster with the largest duration amongst these four clusters (G3, $n = 73$) exhibits an almost linear trajectory over time. On the other hand, cluster G1 with 71 patients presents a more irregular trajectory, with periods of increase and decrease of the ALSFRS-R score at different speeds. Finally, the trajectories of the clusters G4 and G2, with 75 and 72 patients, respectively, are very similar, starting with a rapid decline, and stabilizing around appointment 2. These four trajectories strongly overlap over time, therefore, MoGP does not allow the coherent characterization of the patients within each cluster.

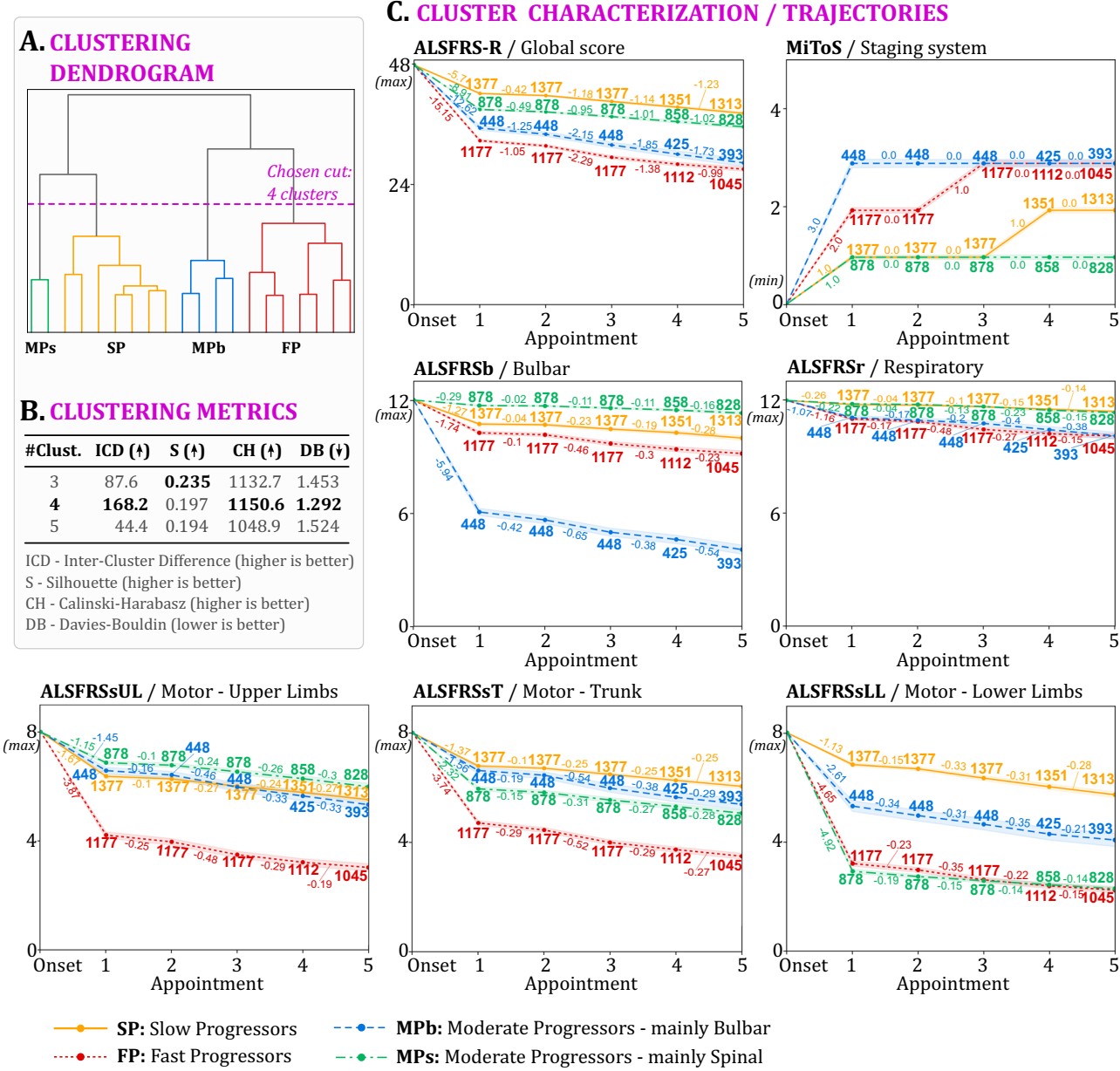

**Fig. 2 | Cluster analysis and characterization on PRO-ACT ALS cohort.**
**A** Dendrogram resulting from ClusTric; (**B**) clustering evaluation scores obtained with 3, 4 and 5 clusters; (**C**) average temporal feature trajectories (lines) and 95% confidence intervals (shades around lines). The numbers next to each point in the trajectories represent the number of patients in the <cluster,appointment> set, whereas the numbers between consecutive appointments indicate the average slope between consecutive measurements in a cluster. Source data are provided as a Source Data file.

In contrast, our approach identified four clusters using the same set of patients, matching those previously found (SP, MPb, MPs and FP). The larger cluster (MPb) comprised 338 patients, while the smaller cluster (FP) comprised 66 patients. Figure 3C illustrates the trajectories of the ClusTric groups. Notably, the trajectories of the four groups demonstrate significant differences. The cluster with 318 patients (SP) displays an almost quadratic trajectory characterized by a gradual decline. This cluster exhibits the longest duration among the groups and matches the SP cluster of the previous analysis. The clusters with 338 (MPb) and 232 (MPs) patients share similarities; however, the latter exhibits a slower decline and a shorter follow-up duration compared to the former. Conversely, the cluster with fewer patients (FP) shows a more irregular trajectory over time. It demonstrates periods of steep decrease followed by periods of moderate decline,

resulting in varying progressions in the ALSFRS-R score. This irregularity may potentially be attributed to the limited number of patients available for analysis over time.

The comparison of the two approaches applied to the PRO-ACT dataset yielded similar conclusions (Fig. 3D–F). However, MoGP identified 92 overlapping clusters (more than in the Lisbon Clinic dataset), contrasting with the 4 different progression groups found by ClusTric.

On the other hand, when comparing the trajectories found by ClusTric in the Lisbon and PRO-ACT datasets, two differences strikeout: in the PRO-ACT dataset, the FP cluster decreases less sharply, while the MPb and MPs clusters have more similar ALSFRS-R trajectories. As expected, patients with isolated bulbar dysfunction would not be recruited in clinical trials, some of these patients can progress without disease spreading for some time, giving this group a slower

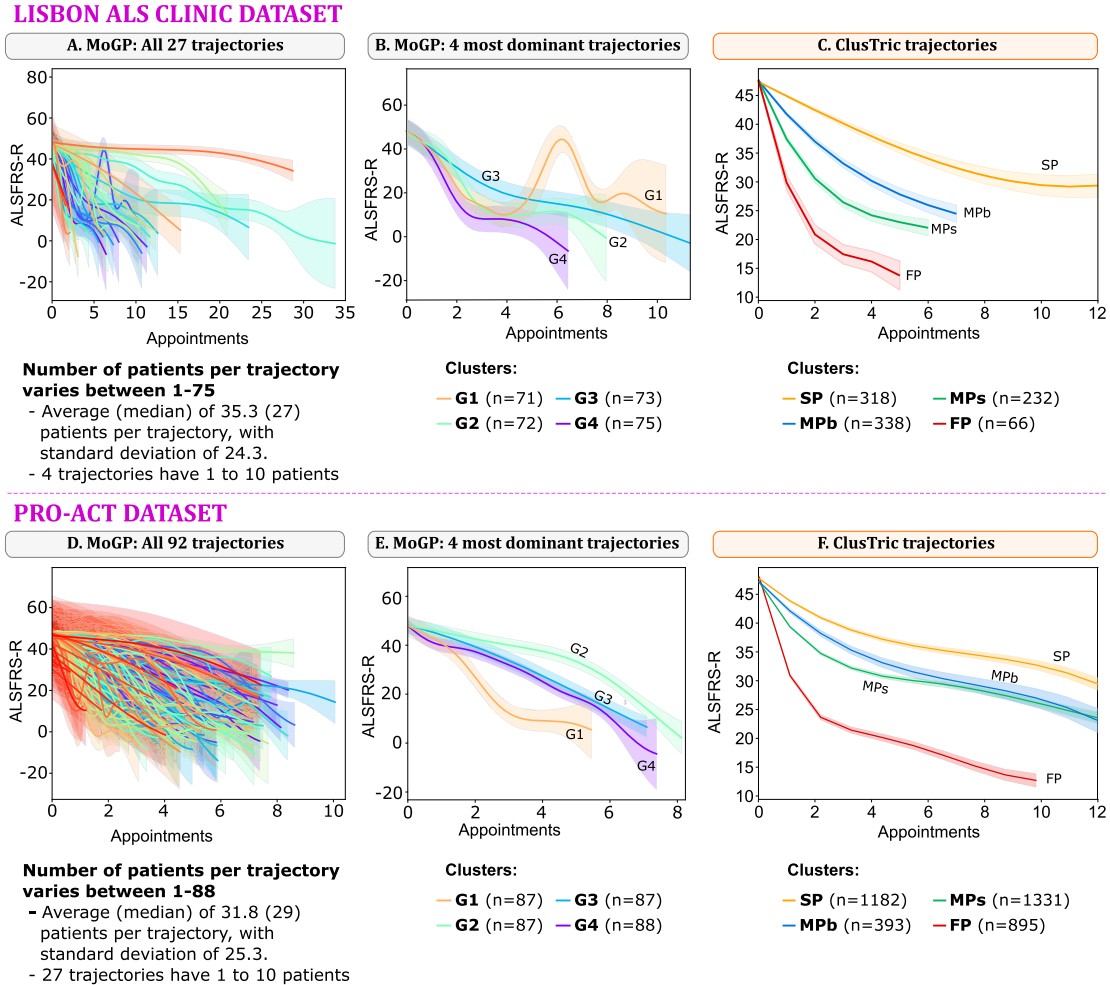

**Fig. 3 | Comparison of ALSFRS-R trajectories per cluster between ClusTric and MoGP.** Top subfigures (**A**–**C**) show the results on the Lisbon ALS dataset, while bottom ones (**D**–**F**) present the results for the PRO-ACT dataset. **A** and **D** show all MOGP trajectories, with **B** and **E** highlighting the four most dominant MOGP trajectories; **C** and **F** show the ClusTric trajectories. Lines represent average values and shaded areas the corresponding 95% confidence interval. For this analysis we pre-processed both cohorts following the steps needed by MoGP[9] on both methods

(which is more restrictive than ClusTric), resulting in a total of 954 and 3801 included patients in the Lisbon and PRO-ACT cohorts, respectively. MoGP identified 27 clusters in the Lisbon cohort (**A**), while ClusTric identified 4 (**C**). Regarding the PRO-ACT dataset, MoGP identified 92 clusters (**D**), while ClusTric identified 4 clusters (**F**). MoGP does extensive smoothing on trajectories resulting in *yy* axis values out of scale.

ALSFRS-R score decay, an observation not disclosed in bulbar-onset patients included in trials.

Furthermore, we evaluated the clustering metrics of both models, MoGP and ClusTric (see Table 4). MoGP yielded negative Silhouette scores, indicating suboptimal partitioning of the data into meaningful clusters. In contrast, ClusTric demonstrated a more effective grouping of the patients.

In conclusion, MoGP identifies trajectories solely based on a single feature, in this case, the ALSFRS-R total score. In contrast, our approach can uncover relationships between multiple features and patients over time. Additionally, it uncovers comprehensive patterns while relying less on exhaustive preprocessing and being approximately 367 times faster than MoGP in this experiment.

**Survival analysis in the Lisbon ALS clinic dataset**
To validate the predictive utility of the proposed stratification we performed a survival analysis on the Lisbon ALS Clinic dataset and compared the results with the 4 most dominant clusters obtained by MoGP.

Survival analysis was done using the Kaplan-Meier method and the log-rank test was used to test the difference between survival curves.

The study was conducted by considering an 8-year follow-up and reflects the duration of the date of the first visit to death, in years. Survival rates are expressed as the percentage of patients surviving for 2 years calculated using the Kaplan-Meier method. All the analyses were carried out in SPSS version 29, considering a statistical significance of 0.05.

Figure 4 presents the result of the survival analysis performed on the Lisbon ALS Clinic dataset. For ClusTric, the survival curve shows that the group FP was the one with the shortest survival rate of 24.6 months (ranging from 19 to 30 months), and a 2-year survival rate of 27% (95% CI of 16–39%). The SP group is the one with the longest survival rate, with a 2-year survival rate of 81% (95% CI of 77–85%) and a mean survival of 53.9 months (ranging from 50 to 57 months). The two moderate progressors groups have similar survival rates and mean survival time. This was confirmed by the pairwise statistical comparison, $p = 0.955$. All other survival curves were considered statistically different (see Table S3 of the Supplementary material).

Regarding MoGP, it identified 27 clusters in the Lisbon cohort, however, only the four most predominant clusters were considered for the survival analysis. The analysis showed that the three groups are quite similar and only group G3 diverges from the remaining. In

**Table 4 | Clustering evaluation**

| | | Silhouette (↑) | CH (↑) | DB (↓) |
|---|---|---|---|---|
| Lisbon ALS clinic | MoGP (all clusters) | −0.437 | 15.682 | 10.929 |
| | MoGP (4 dominant clusters) | −0.057 | 2.921 | 10.064 |
| | ClusTric | **0.237** | **310.179** | **1.373** |
| PRO-ACT | MoGP (all clusters) | −0.701 | 1.277 | 49.077 |
| | MoGP (4 dominant clusters) | −0.067 | 2.018 | 21.485 |
| | ClusTric | **0.204** | **1085.275** | **1.377** |

Silhouette, Calinski-Harabasz (CH), and Davies-Bouldin (DB) scores obtained with the two models, MoGP and ClusTric, on the Lisbon and PRO-ACT datasets. For ClusTric, the scores were calculated using the similarity matrices produced by the triclustering algorithm; for MoGP the scores were calculated using the first 3 ALSFRS-R scores of each patient. The rationale is that ClusTric considers relations between features, while MoGP only uses ALSFRS-R.
The best results are in bold.

particular, group G3 presented statistically significant differences when compared to any other group (see Table S3 of the Supplementary material). The mean survival time and the 2-year survival rate of group G3 are higher than in the remaining groups (Fig. 4).

### Prediction of group progression

Despite the specific disease progression identified by the proposed method, it is possible that, at a given moment, the progression changes for certain patients. Hence, we set to find out whether the patients always remained within the original cluster, or changed over time.

To answer this question, we investigate the progression of the group assignment for each patient from appointments 1–3 (approximately, the first 6 months) to appointments 3–5 (approximately, the following 6 months). For this, we employed a Random Forests classifier, using the clustering labels obtained from the previous analysis as the target prediction. The classifier was trained using the transformed data of the initial three appointments. We then predicted the patient's cluster considering the third to fifth appointments. The analysis is depicted in Fig. 5A indicates that the majority of patients (66.82%) remained in the same cluster during the 12 months of follow-up, whereas 33.18% of patients changed their cluster.

Given the lack of ground truth to confirm the assignments obtained by the classifier, we decided to do a survival analysis of the patients belonging to one group (e.g., SP) found by the ClusTric method and were then classified in the second period of follow-up as belonging to different groups. Figure 5B presents the analysis for the patients coming from group SP (composed of 313 patients) that were later classified either as SP, MPb, and MPs. The new SP group is composed of 196 patients and is the one with a higher survival rate, with a 2-year survival rate of 93% (95% CI of 90–97%). The MPb in the second period is composed of 47 patients and has the lowest survival rate, in particular, a 2-year survival rate of 78% (95% CI of 66–90%). Finally, MPs is composed of 69 patients and has a 2-year survival rate of 83% (95% CI of 74–92%).

Moreover, from the patient transition table in Fig. 5A, we notice that some transitions correspond to outliers, e.g., one patient shifted from SP to FP. Although a steep decline of 20 points in the ALSFRS-R score was observed for that patient in the second period, further conclusions could not be drawn from a single observation.

Finally, the survival curves for the remaining transitions are not presented due to a small sample size. In particular, the transition from MPb to MPs includes only 27 patients, and the transition from MPs to FP has only 10 patients. Nonetheless, when analyzing the 2-year survival rate for such transitions, we noticed that patients who remain in the MPb group (a total of 130 patients) have a survival rate of 74% (95% CI of 66–81%), and the ones that shifted to FP coming from MPb (a total of 61 patients) have a survival rate of 51% (95% CI of 38–64%).

## Discussion

ClusTric, the proposed temporal stratification approach combining triclustering and hierarchical clustering, identified four distinct clusters in ALS patients, each with unique disease progression patterns. The results (Figs. 1 and 2) confirmed that ALS disease trajectories are heterogeneous and non-linear. These four clusters differ according to onset features and disease progression in several functional domains:

- Cluster SP shows an overall slow progression (Figs. 1C and 2C). In the Lisbon ALS Clinic dataset, this is evident in the bulbar and respiratory domains. However, this cluster has similarities to cluster MPb in the upper limbs and trunk domains and has a slightly worse progression than cluster MPb in the lower limbs domain. Cluster SP is composed mostly of males (64.2%) and younger patients (mean onset age of 58.92 years) at disease onset (see Table 3). Due to the slow progression, this cluster has higher VC and ALSFRS-R at 1st visit (96.94 and 43.21, respectively), lower ALSFRS-R decay in the first 6 months of the disease (decrease of 0.58/month), and larger amplitude of the diaphragm motor response (0.67mV). These patients have a diagnostic delay of 17.35 months on average.

- Cluster MPb comprises patients with moderate progression, showing better limb and trunk functionality (Figs. 1C and 2C). In the Lisbon ALS Clinic dataset, this cluster has a higher proportion of females (51.9%) and a higher mean age at onset (64.5 years old) than cluster SP, as expected for the bulbar-onset phenotype. These patients have a diagnostic delay of 13.17 months on average. The VC and ALSFRS-R scores at 1st visit and the ALSFRS-R decay in the first 6 months of the disease are worse than those of cluster SP, but still slightly better than those of cluster MPs.

- Cluster MPs includes moderate progressors, but with better bulbar function than MPb (Figs. 1C and 2C). The patients from the Lisbon ALS Clinic dataset integrating this cluster have a similar respiratory function to those comprising cluster MPb, but a limb and trunk score pattern similar to cluster FP. They are predominantly male (59.8%) with an age at onset of 60.04 years on average. The baseline diaphragm motor response amplitude is lower than in cluster SP, but slightly higher than in cluster MPb.

- Cluster FP includes patients with the poorest prognosis (Figs. 1C and 2C). In the Lisbon ALS Clinic dataset, this cluster has the highest proportion of female patients (only 29.9% are male), and the highest age at onset (69.73 years on average). Values at 1st visit of VC, ALSFRS-R scores and diaphragm motor response amplitude are the lowest, and the ALSFRS-R score decay rate is the highest in the first 6 months of the disease. On the other hand, in the PRO-ACT dataset, this cluster is mostly composed of young men (62%), with an age at onset of 55 years on average (see Table S2 of Supplementary material).

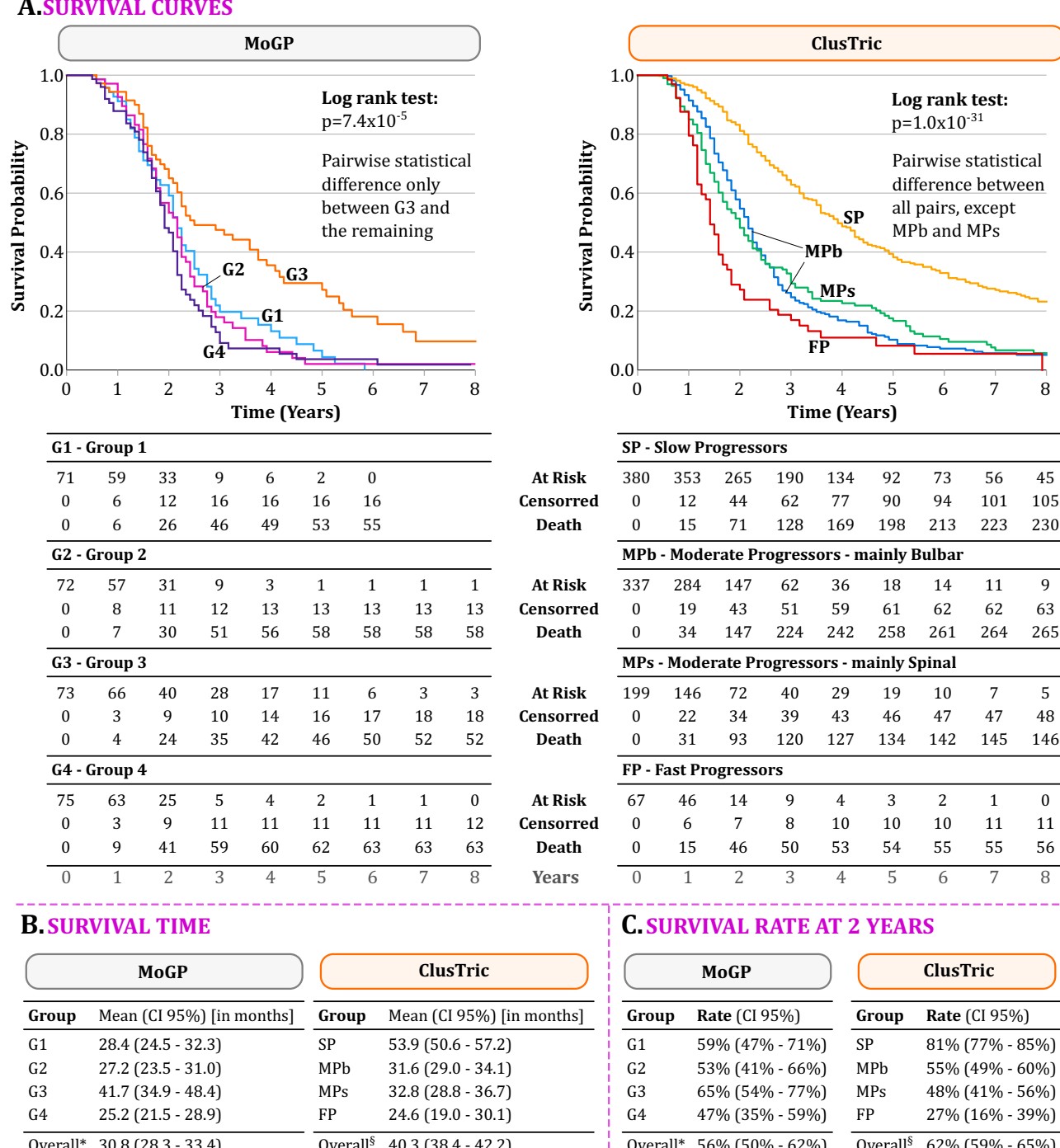

**Fig. 4 | Survival analysis on the Lisbon ALS cohort. A** Kaplan–Meier curves showing up to 8 years follow-up for MoGP and ClusTric comparing the groups identified by each method and tables with the number of patients at risk, censored, and that died by year in each group. Log-rank test between survival curves and *p*-values of pairwise comparisons between curves in Table S3 of the Supplementary material. **B** Mean survival time (in months) and 95% of confidence interval for MoGP and ClusTric. **C** Survival rates as the percentage of patients surviving for 2 years calculated using the Kaplan–Meier method.

Our results underline the differences between clinical trial-based vs hospital-based ALS patients. Clinical trial population is characterized by young-onset, preserved respiratory function, more men, and predominant spinal-onset phenotype. In addition, patients enrolled in clinical trials have, in general, a significantly better prognosis than those of a hospital-based population[29]. In particular, patients enrolled in PRO-ACT have higher body mass index and more frequent upper limb weakness with a slower disease spreading (a phenotype more common in young men[30]). Patients with bulbar-onset and without other regions affected are not candidates for clinical trials, although

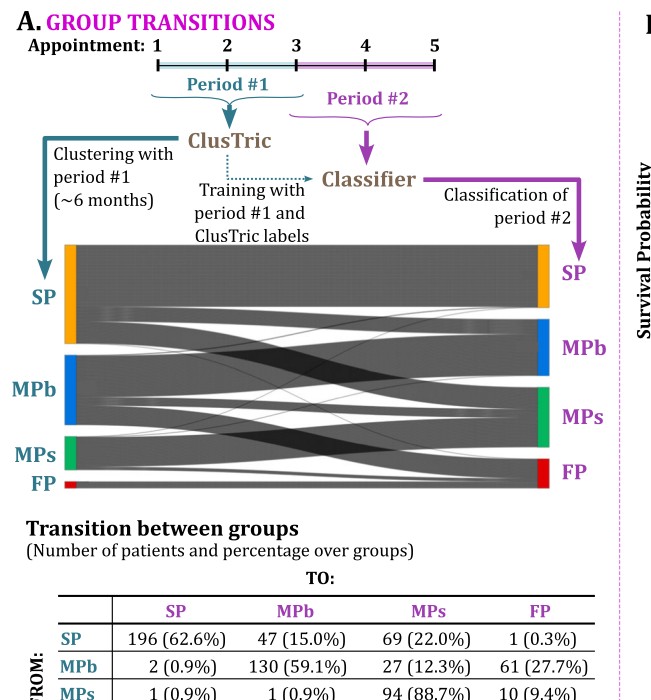

**A. GROUP TRANSITIONS**

**B. SURVIVAL OF PATIENTS COMING FROM SP GROUP (Period #1)**
(Lines represent different groups in period #2)

**Transition between groups**
(Number of patients and percentage over groups)

|  | | TO: | | |
|---|---|---|---|---|
| | **SP** | **MPb** | **MPs** | **FP** |
| **SP** | 196 (62.6%) | 47 (15.0%) | 69 (22.0%) | 1 (0.3%) |
| **MPb** | 2 (0.9%) | 130 (59.1%) | 27 (12.3%) | 61 (27.7%) |
| **MPs** | 1 (0.9%) | 1 (0.9%) | 94 (88.7%) | 10 (9.4%) |
| **FP** | 0 (0.0%) | 0 (0.0%) | 0 (0.0%) | 21 (100%) |

(FROM: appears as row label on left)

**Survival rate at 2 Years**
SP at period #1

| | | Survival Rate (CI 95%) |
|---|---|---|
| Group in period #2 | **SP** | 93% (90% - 97%) |
| | **MPb** | 78% (66% - 90%) |
| | **MPs** | 83% (74% - 92%) |

**Fig. 5 | ClusTric clusters transition from 6 to 12 months follow up on the Lisbon ALS cohort. A** Sankey diagram with patient's transitions between ClusTric clusters and the number of patients that transition from one cluster to any other cluster. **B** Survival analysis of patients coming from group SP in the first 6 months to other progression groups (namely, SP, MPb, and MPs) in the following 6 months. Log-rank test between survival curves and *p*-values of pairwise comparisons between curves are in Table S4 of Supplementary material.

some of these patients progress slowly to other regions (slower spreading). This supports a higher ALSFRS-R observed in hospital-based patients when compared to PRO-ACT bulbar-onset patients, representing two different populations of bulbar-onset patients. Moreover, the hospital-based Lisbon population is more hetero-geneous, including older patients with axial-onset, not eligible for clinical trials due to the poor respiratory tests associated with this phenotype.

The characterization of the clusters obtained by ClusTric provides valuable insights into different disease progression patterns amongst ALS patients. In particular, the survival analysis performed for each group shows that:

- Patients in cluster SP have longer survival on average.
- Clusters MPb and MPs do not reveal statistical differences regarding survival, but their expected survival is longer than in cluster FP and shorter than in cluster SP.
- Patients in cluster FP have the shortest survival (24 months on average).

In the transition study, we predict the progression group for each patient in appointments 3–5 training a classifier using the ClusTric results as the class label, assuming that the group does not change throughout disease progression. The results in the Lisbon ALS clinical dataset show that a significant number of patients (66.8%) do not change progression group (refer to Fig. 5), while some changes occur:

- 12.3% of the patients in cluster MPb transitioned to cluster MPs and 27.7% to cluster FP. Only a residual percentage (0.9%, corre-sponding to 2 patients) transitioned to cluster SP.
- 15.0% of the patients in cluster SP transitioned to cluster MPb and 22.0% to cluster MPs. Only one patient transitioned to cluster FP.
- 9.4% of the patients in cluster MPs transitioned to cluster FP, while only 0.9% transitioned to clusters MPb and SP (1 patient, respectively).

- All patients in cluster FP (with the worst prognosis) remained in the same cluster.

This transition study shows that fast progressors have an invariably poor prognosis. On the other hand, slow progressors can later show fast progression and a worse prognosis.

The transition between groups ("classification errors") reflects two realities: (i) typical classification errors; or (ii) actual changes in the patient's disease progression, corresponding to a "classifier error" that in reality is not an error since the patient actually evolves differently. It is well-known in clinical practice that a small number of patients deviate from the expected disease progression[31]. Their disease pro-gression is frequently non-linear, limiting the accuracy of predictive models, since some of these patients can progress more or less rapidly than previewed[9,32]. In this context, increasing the number of clusters in ClusTric would not capture these group changes since the disease progression of these group-changing patients is coherent with that of the group in the time window used for clustering. The clustering algorithm could only capture these group changes if more appoint-ments were considered to compute the clusters.

Our methodology is flexible in using any number of appointments and incorporating static features. In this work, we used three con-secutive appointments to study ALS disease progression following clinical guidance. The inclusion of static features resulted in less coherent groups since the higher patients' heterogeneity reflected in these features is not translated into the same heterogeneity when analyzing disease progression using the functional scores (see sec-tion 1.1 of Supplementary Material). We note that the number of assessments to be used by ClusTric and the inclusion or absence of static features should be settled according to the specificities of each disease and stratification problem. In particular, in the case of strati-fication targeting the enrollment in clinical trials only one assessment may be required, ClusTric can also be applied (as shown in section 1.2

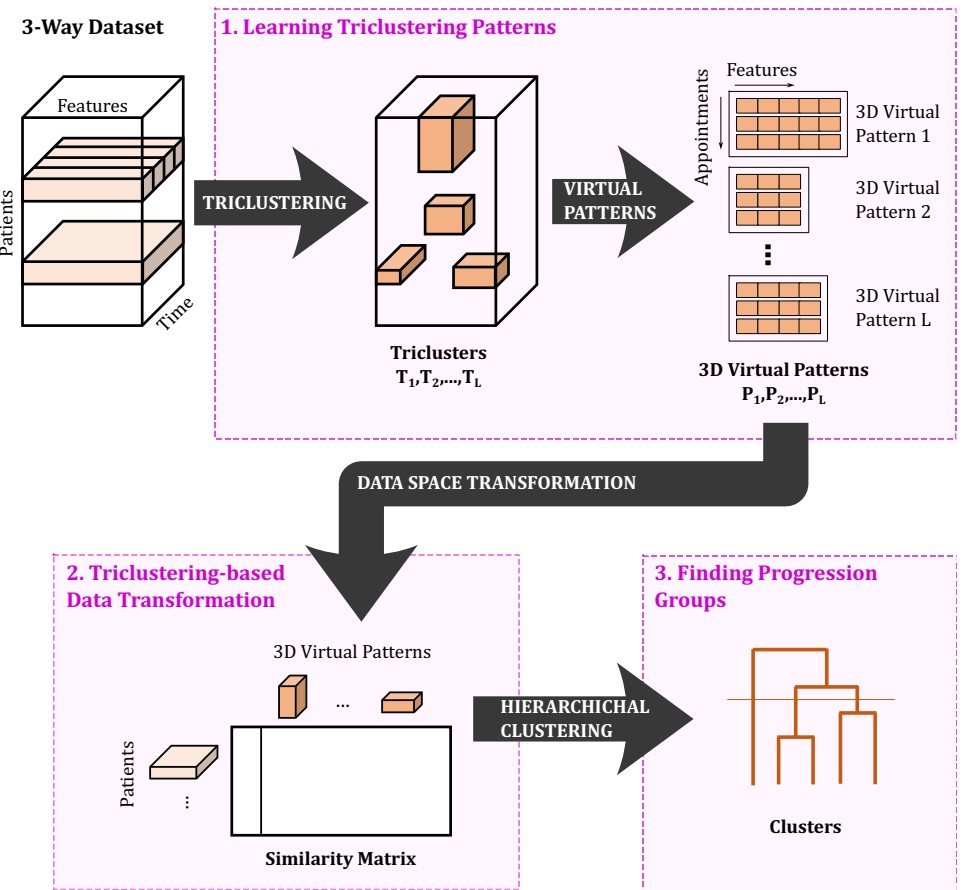

**Fig. 6 | ClusTric method.** The proposed method is three-fold: (1) learning triclustering patterns; (2) triclustering-based data transformation; (3) finding progression groups.

of Supplementary Material). As expected, when only one appointment is used, the obtained groups reflect the differences in the patients' functional state (with specific domains affected) and do not capture disease progression.

Our results show that ClusTric, a temporal stratification methodology combining triclustering and agglomerative clustering, was effective in capturing ALS disease progression by grouping a heterogeneous hospital-based population in four very distinct prognosis groups validated by clinicians. Furthermore, this model results in a lower number of groups than other approaches[20,33] using a simple and easily available set of clinical information. We encourage the application of our method in other ALS populations to establish its utility.

## Methods

We propose an approach to learn disease progression groups called ClusTric. This process is 3-fold: (1) finds coherent disease progression patterns using triclustering; (2) computes the representative pattern of the resulted triclusters; (3) and then clusters the patients according to their similarity to these patterns using hierarchical clustering. Figure 6 overviews the proposed method. Although this method is herein applied to ALS stratification, it can be used as a standalone approach to understanding the progression of other diseases (e.g., multiple sclerosis or Alzheimer's disease). The remaining of this section details each step of the proposed method.

### Learning triclustering patterns

We start by explaining some key concepts to better understand the process of finding disease progression patterns with triclustering[25,34]. In particular, we start with the definition of a three-way dataset.

**Definition 1.** (Three-way dataset) A three-way dataset $D = \{d_{ijk}\}$ is defined by set of subjects $X = \{x_1, \ldots, x_{|X|}\}$, features $Y = \{y_1, \ldots, y_{|Y|}\}$, and contexts $Z = \{z_1, \ldots, z_{|Z|}\}$, where the elements $d_{ijk}$ relate subject $x_i$, feature $y_j$, and context $z_k$.

In the context of our work, we consider heterogeneous data that is characterized by the presence of $N$ subjects described by a set of static features $Y_{static}$; and temporal features $Y_{temporal}$, associated with a three-way temporal dataset, where elements $d_{ijk}$ relate subject $x_i$, feature $y_j \in Y_{temporal}$, and time point $z_k$. The number of time points $|Z|$ can depend on the temporal study and be set to 1 if relevant.

Given a three-way dataset, a tricluster and a triclustering solution are defined as follows:

**Definition 2.** (Tricluster) Given D, a tricluster $\mathcal{T} = (I, J, K)$, is a subspace defined by $I \subseteq X$ subjects, $J \subseteq Y$ features, and $K \subseteq Z$ contexts, where $d_{ijk}$ denotes the elements of $\mathcal{T}$.

**Definition 3.** (Triclustering Solution) Given $D$, the triclustering task aims to find a set of triclusters $\{\mathcal{T}_1, \ldots, \mathcal{T}_L\}$ such that each tricluster $\mathcal{T}_l$ satisfies predefined criteria of homogeneity and statistical significance[25].

In the particular case of ClusTric, we employ the `TCtriCluster`[26] algorithm to find temporally coherent and contiguous triclusters. This algorithm is an extension of `triCluster`[35], a quasi-exhaustive approach, able to mine arbitrarily positioned and overlapping triclusters with constant, scaling, and shifting patterns from three-way data. It encompasses three steps: (1) creating a multigraph that ensures comparable value ranges among all sample pairs, (2) identifying maximal biclusters from the multigraph, which is formed for each time

point (representing slices of the three-way dataset), and (3) combining similar biclusters from different time points to extract triclusters.

After obtaining the triclustering solution, the approach computes the 3D Virtual Pattern for each tricluster (progression pattern):

**Definition 4.** (3D Virtual Pattern) Given a tricluster $\mathcal{T}_l$, its corresponding virtual pattern $\mathcal{P}_l$ is defined as a set of $|K|$ sets, $\mathcal{P}_l = \{\rho_k | k \in K\}$, where each set $\rho_k$ with size $|J|$ is a pattern composed of the average (or the mode, in case of categorical features) of values across subjects $I$ for each feature $J$:

$$\rho_k = \left\{ \frac{1}{|I|} \sum_{i=1}^{|I|} d_{ijk} \,\middle|\, j \in J \right\}. \tag{1}$$

### Triclustering-based Data Transformation

After obtaining the $L$ 3D Virtual Patterns, a $|X| \times (|K| \times L)$ similarity matrix between patients and patterns is computed using the Euclidean distance (other distances can be used) as follows:

**Definition 5.** (Euclidean-based Distance) Given a 3D virtual pattern $\mathcal{P}_l$, and the set of values in the $J$ features of a subject $x_i$ in a context $z_k$, $F_k(x_i) = \{d_{i1k}, d_{i2k}, \ldots, d_{i|J|k}\}$, its Euclidean-based distance is computed by:

$$S(\mathcal{P}_l, F_k(x_i)) = \left\{ \sqrt{\sum_{e=1}^{|J|} (F_k(x_i)_e - \rho_{k_e})^2} \,\middle|\, \rho_k \in \mathcal{P}_l \right\}. \tag{2}$$

This similarity matrix between subjects and temporal patterns can then be used to compute the disease progression groups. A similar approach was used in refs. 23,26 with promising results in supervised learning.

Static features can also be considered directly in the similarity matrix or by using 2D patterns discovered by biclustering as performed in ref. 23. In the first case, the similarity metric might need to be adapted according to the features' types.

### Finding progression groups

Given the similarity matrix $S$ previously computed, ClusTric groups subjects using an Agglomerative Hierarchical clustering algorithm, such as Ward linkage and Euclidean distance as the affinity metric.

Ward's linkage, a variance-based method, was chosen for its ability to minimize variance within clusters, making it well-suited for handling noisy data, such as longitudinal data from patients with progressive diseases.

An advantage of using a hierarchical clustering algorithm is not having to choose the number of clusters a priori. This can be achieved by manually inspecting the dendrogram, which provides intuitive insights into the optimal number of clusters, or by considering internal clustering validation metrics, such as the silhouette index.

### Reporting summary

Further information on research design is available in the Nature Portfolio Reporting Summary linked to this article.

## Data availability

The Lisbon ALS Clinic dataset analyzed during the current study is available under restricted access to ensure patients' rights to privacy and anonymity; access may be obtained by contacting Prof. Mamede de Carvalho (Instituto de Medicina Molecular - Faculdade de Medicina Universidade de Lisboa; mamedealves@medicina.ulisboa.pt; expected response time: 1 month). PRO-ACT database is available for download upon registration at https://alsdatabse.org. Source data are provided with this paper for Figs. 1, 2, and S1–S4 of the Supplementary material. Other source data are not publicly available due to the inclusion of private patient-level clinical information. For reproducibility, access may be obtained by contacting corresponding authors (expected response time: 1 month). Source data are provided with this paper.

## Code availability

The ClusTric source code was made available at: https://github.com/LxMLearners/ClusTric[36].

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

## Acknowledgements

Data used in the preparation of this article were partially obtained from the Pooled Resource Open-Access ALS Clinical Trials (PRO-ACT) Database. The following organizations and individuals within the PRO-ACT Consortium contributed to the design and implementation of the PRO-ACT Database and/or provided data: ALS Therapy Alliance, Cytokinetics, Inc., Amylyx Pharmaceuticals, Inc., Knopp Biosciences, Neuraltus Pharmaceuticals, Inc., Neurological Clinical Research Institute MGH, Northeast ALS Consortium, Novartis, Prize4Life Israel, Regeneron Pharmaceuticals, Inc., Sanofi, Teva Pharmaceutical, Industries, Ltd., The ALS Association. None of the organizations had any influence on the design, conceptualization, or implementation of the methodology, performing the experiments, analysis of the result, writing of the manuscript, or the decision to submit it for publication. This work was partially supported by Fundação para a Ciência e a Tecnologia (FCT) through project AIpALS ref. PTDC/CCI-CIF/4613/2020 (https://doi.org/10.54499/PTDC/CCI-CIF/4613/2020) to D.M.A., D.F.S., M.G., M.dC., S.C.M., and H.A.; LASIGE Research Unit, ref. UIDB/00408/2020 (10.54499/UIDB/00408/2020) and ref. UIDP/00408/2020 (https://doi.org/10.54499/UIDP/00408/2020) to D.M.A., D.F.S., S.C.M., and H.A.; INESC-ID Research Unit, ref. UIDB/50021/2020 (https://doi.org/10.54499/UIDB/50021/2020) to P.T.; and a PhD research scholarship ref. 2020.05100.BD (https://doi.org/10.54499/2020.05100.BD) to D.F.S.; and by the BRAINTEASER project which has received funding from the European Union's Horizon 2020 research and innovation program under the grant agreement No 101017598 (https://doi.org/10.3030/101017598) to M.G., M.dC., S.C.M., and H.A.

## Author contributions

D.M.A., D.F.S., S.C.M., and H.A. conceptualized the study and designed the methodology; D.F.S. implemented the triclustering framework; D.M.A. implemented the ClusTric tool and performed the experiments; D.F.S., P.T., H.A., and S.C.M. critically revised the results; D.M.A., D.F.S., P.T., and H.A. drafted the manuscript; S.C.M., M.G., and M.dC. revised the manuscript; S.C.M. acquired funding for research; M.dC. and M.G. collected and processed the patients' follow-up Lisbon Clinic data and revised the study from the clinical point-of-view; S.C.M., P.T., and H.A. supervised the work.

## Competing interests

The authors declare no competing interests.

## Ethical approval

The study was conducted in accordance with the Declaration of Helsinki and was approved by the local (Faculty of Medicine, University of Lisbon) ethics committee.

## Informed consent

Informed consent to participate in the study was obtained from all participants. Access to Lisbon ALS Clinic data was granted in the context of project AIpALS (PTDC/CCI-CIF/4613/2020), where the authors' institutions participate.
