## [Peer Review File · Nature Communications]

REVIEWER COMMENTS

Reviewer #1 (Remarks to the Author):

Amaral et al. present a novel approach to find different progression patterns in neurodegenerative diseases such as ALS. The method - ClusTric, is based on triclustering to identify different progression patterns, followed by agglomerative clustering to match patients to different progression patterns. One advantage of this method is that it can work with multivariate time-series data to mine diverse disease progression patterns in an unsupervised way. The analysis identifies 4 different clusters of ALS patients from the Lisbon ALS clinic dataset that follow different patterns of progression. Across these clusters, there are differences such as patient state at baseline visits and rate of decline. The analysis also compares the results to well-established staging systems such as MiToS and other disease progression models such as MoGP.

Despite the strengths, I think that there are some important issues with the current form of this work. These are –

1. It is not very clear if the 4 different clusters, i.e. SP, MPb, MPm, and FP, represent different progression patterns or just different disease stages? Disease progression patterns are related to disease subtypes which is different from disease stage. These are two different ideas while modeling disease progression, but the results do not show a disentangled picture between them. Results in 4a show how the cluster assignments change to more severe forms over time. Figure 5 shows a correlation between the MiToS stage and the clusters. This casts a doubt on whether the clusters are largely representative of the disease stage? Can the method be extended to output a unique combination of disease subtype and disease stage?

2. Lack of replication across other datasets – It would be interesting to see the method being applied to an external validation dataset. Does another ALS dataset also show 4 clusters emerge from it?

3. Prognostic utility of the assigned clusters – The work can benefit from a survival analysis that uses the assigned clusters.

4. Comparison with MoGP – I feel the comparison between the two methods can be extended to include their prognostic utility. Which of the two methods give better results in predicting future decline?

Minor –

I feel the method section can be improved to present a clear mathematical framework of the ClusTric algorithm. This helps the reader in getting a clear picture of the method.

Reviewer #2 (Remarks to the Author):

The authors modify a previously established tri-clustering algorithm as their base framework for stratifying ALS patient progression based patient-record-time. They apply the unsupervised framework to a single cohort from the Lisbon ALS clinic and describe the resultant characteristics of the four primary clusters and transitions between clusters. They then compare their method to a recently published Gaussian mixture model for ALS progression and stratification.

The work is interesting and significant. It does correctly characterize the non-linearities in ALS that have long been known even though many works often have forced linear assumptions. While the work is interesting and significant, it is not entirely original from a machine learning standpoint given is primarily an application of an existing method. It is original from a domain standpoint given prior work has not used the method in ALS.

The authors explained well the clustering method. The authors correctly state there is not explicit ground truth validation available given this is an unsupervised method. However, there is more the authors could do to assess their results. The biggest drawback to the present manuscript's results is that there is only one cohort assessed. The authors really need to try this method on multiple cohorts and compare them. Comparing the features of the cohort and then comparing the resulting clustering results would provide much more clarity and trust into how well this algorithm works. Using only one cohort does not allow the algorithm to be fairly assessed. There are other open access data available that the authors could use to provide critical baseline assessments. In a journal of this impact, I would consider adding at least one and preferably 2 additional cohorts as a necessity to evaluating the work.

The lack of inclusion of static features is another concern that will need fixed in order to maximize the impact of the present work. It appears the authors did perform a post-hoc analysis of static features after clustering. However, it would be better if there was a more explicit framework for inclusion of static variables. It is possible the authors did have such a framework but, if so, it was not clear based on the currency manuscript.

While the stratification here is based on progression, it would improve the clinical utility of the work if survival were more overtly incorporated. The authors should provide a censored graph of survival

probability over time. Then, they should illustrate survival probability for each of their clusters graphically and then do the follow-up statistical analysis.

I think it is correct for the authors to compare to the recent Gaussian mixture model. However, the authors over sell their results in saying their clustering model is "better" or "faster" without actually providing explicit evidence of these claims. The language of these claims needs to be more appropriate.

Reviewer #3 (Remarks to the Author):

This manuscript addresses an important area of need in the field of ALS- developing tools to better predict disease progression and stratify patients for clinical trials. The method of combining triclustering and hierarchical clustering to better account for non-linear progression trajectories is innovative and well-described in this manuscript, but will be further assessed by statistical review. This review will focus on the ALS clinician-scientist perspective. There are a few major concerns with this manuscript in its current form.

1) The authors inappropriately interchange ALS staging systems (static measurements of current disease stage) with ALS projection trajectories (rates of functional decline over time). Staging systems such as MITOS give no information about rate or overall trajectory of progression- the expectation is that each individual person with ALS will progress sequentially through each of the MITOS stages over the course of their disease, but at different rates. In other words, both fast progressors and slow progressors will move through the same MITOS stages in the same order, but the fast progressors move through each stage at a faster rate. It could make sense to compare time spent in various MITOS stages with a progression trajectory, but the current analysis that compares a static MITOS stage with a progression trajectory is not informative.

2) The "Related Work" section appropriately points out the flaws with linear ALS models, but selectively leaves out other more sophisticated modeling approaches that have been more successful. In particular, this section should cite and discuss the work by Berry et. al. "Identifying patterns in amyotrophic lateral sclerosis progression from sparse longitudinal data" and the work of Westeneng et. al. "Prognosis for patients with amyotrophic lateral sclerosis: development and validation of a personalised prediction model."

3) The authors do cite the work of Ramamoorthy and colleagues and compare performance of both models. However, they compare the performance of their 4 proposed trajectories with 27 different

trajectories from the Ramamoorthy paper. The Ramamoorthy paper also goes on to identify 4 dominant clinical progression patterns that are not acknowledged in this manuscript, when comparison of both 4-trajectory models would be a more informative comparison.

Response to Reviewers

General Comments to Reviewers:

Thank you for all your attention, detailed feedback, and for providing us the opportunity to resubmit an improved version of our manuscript, **NCOMMS-23-40504-T**. This document details how the raised concerns were incorporated into the revised version of the manuscript.

The major changes to our initial submission are the following:

- The whole manuscript was restructured to match the journal guidelines. So, most of the sections changed names and were reordered. The new structure is:
 - Abstract
 - Introduction (includes the previous Related Work section)
 - Results
 - ALS Datasets
 - ClusTric in Lisbon ALS Clinic Dataset
 - ClusTric in PRO-ACT Dataset
 - Comparison with MoGP
 - Survival Analysis in the Lisbon ALS Clinic Dataset
 - Prediction of Group Progression
 - Discussion
 - Methods
 - Learning Triclustering Patterns
 - Triclustering-based Data Transformation
 - Finding Progression Groups
 - Data Availability
 - Code Availability
 - References
- The proposed stratification method, ClusTric, is now also validated in a publicly available ALS cohort - PRO-ACT (i.e., in addition to the Lisbon ALS Clinic cohort used in the original manuscript). The characteristics of the groups found in this new cohort corroborate the initial assumptions made with the study of the Lisbon ALS clinic cohort.
- A survival analysis was performed to validate the proposed clustering method, compare it with the MoGP framework, and validate the prognostic utility of our stratification model, predicting the time-to-death in each assigned cluster. The results proved promising, proving the strength of the stratification procedure in personalized predictions in ALS. The survival analysis within clusters confirmed the groups' progression, with slow progressors living more than the fast progressors. Moreover, compared to the groups found by the MoGP method, the ClusTric groups show more significative differences in survival curves, survival time, and survival rate at two years.

- We verified how the patients' progression changed 6 months after the considered time window for stratification. This analysis shows that patients either remain in the same group (67%) or progress to a more severe group.

The manuscript was carefully expanded, proof-checked, and revised to address all the received comments.

Finally, we have renamed the group MPm (Moderate Progressors, mainly motor) to MPs (Moderate Progressors, mainly spinal) to be more accurate regarding the disease progression.

The changes to the revision of the original manuscript are highlighted in **blue**.

Reviewer #1:

Amaral et al. present a novel approach to find different progression patterns in neurodegenerative diseases such as ALS. The method - ClusTric, is based on triclustering to identify different progression patterns, followed by agglomerative clustering to match patients to different progression patterns. One advantage of this method is that it can work with multivariate time-series data to mine diverse disease progression patterns in an unsupervised way. The analysis identifies 4 different clusters of ALS patients from the Lisbon ALS clinic dataset that follow different patterns of progression. Across these clusters, there are differences such as patient state at baseline visits and rate of decline. The analysis also compares the results to well-established staging systems such as MiToS and other disease progression models such as MoGP.

Answer 1.0: *We would like to thank the reviewer for the comprehensive review provided and for the positive comments regarding the soundness of our work. Below, we respond to the concerns raised.*

Despite the strengths, I think that there are some important issues with the current form of this work. These are –

1. It is not very clear if the 4 different clusters, i.e. SP, MPb, MPm, and FP, represent different progression patterns or just different disease stages? Disease progression patterns are related to disease subtypes which is different from disease stage. These are two different ideas while modeling disease progression, but the results do not show a disentangled picture between them. Results in 4a show how the cluster assignments change to more severe forms over time. Figure 5 shows a correlation between the MiToS stage and the clusters. This casts a doubt on whether the clusters are largely representative of the disease stage? Can the method be extended to output a unique combination of disease subtype and disease stage?

Answer 1.1: Thank you for this valuable comment regarding the difference between the disease progression patterns and stages. In fact, our method intends to learn from the disease progression patterns and achieve stable groups when looking at a 6-month patient follow-up. Each group is characterized by disease progression patterns, given its composition. The formation of the groups was guided by the 6-month data. This is the timeframe in which we verified how the patients progress. The clusters observed in this analysis represent 4 different types of ALS progressions. This is the output of our stratification model in Lisbon ALS clinic data. The identified groups showed specific disease progressions that are representative of the patients' evolution, which can not be understood as subtypes of the disease since a given patient could start progressing slower or faster at some point according to specific functional domains.

In fact, we verified that using an additional 6 months of follow-up, despite the majority of patients remaining in the same group, some of them moved from one cluster to another, which means that in specific functional domains, their progression changes, making them closer to the progression identified for a different cluster than that of the initial assignment.

Moreover, in agreement with the reviewers' comments, we removed the study that compared the groups with MITOS stages, given that a staging system does not assess progression. However, we added the trajectories of MITOS progression within groups to show that the different groups revealed different transitions between stages over time (**Fig. 1C**).

Those results were then validated by the parallel study done with the PRO-ACT database (**Fig. 2C**).

2. Lack of replication across other datasets – It would be interesting to see the method being applied to an external validation dataset. Does another ALS dataset also show 4 clusters emerge from it?

Answer 1.2: Thank you for your insightful comment. We genuinely appreciate your valuable input and completely agree with the importance of assessing our algorithm across another cohort for a more comprehensive evaluation.

As of our current knowledge, we have actively explored the availability of publicly accessible datasets that closely match the distinctive characteristics of the Lisbon ALS clinic dataset used in our research. Regrettably, we have not identified datasets with precisely mirrored features. The Lisbon dataset, derived from patient follow-ups at the local clinic and encompassing data from all ALS patients followed at the largest hospital in Portugal since 1995, possesses unique characteristics, and thus finding an identical cohort is challenging.

Nevertheless, we understand the significance of diversifying our analysis to enhance the overall robustness of our model. In response to this concern, we conducted a parallel analysis using the PRO-ACT database, a publicly accessible ALS cohort. However, it's crucial to note that the PRO-ACT dataset significantly differs from the Lisbon dataset regarding data specificities (see **Table 1** for the characterization of the population in each cohort). The former originates from ALS clinical trials, where patient selection involves meticulous processes and invitations based on predetermined constraints for trial eligibility,

introducing inherent biases. This was also the cohort used by the authors of the MoGP method with which we compared our results.

Interestingly, despite the relevant differences between the two cohorts, four groups emerged from the analysis in the PRO-ACT dataset that corresponded to the ones identified in the Lisbon cohort (see “**ClusTric in PRO-ACT Dataset**”). However, the cluster MPs exhibits a slower progression in the Upper Limbs subscore (ALSFRSsUL) and a faster progression regarding the remaining motor subscores (ALSFRSsT and ALSFRSsLL). This difference arises because the measurements of the ALSFRSsUL subscore in the PRO-ACT dataset exhibit much smaller variations over the considered timeframe when compared to the previously used Lisbon cohort (see **Table 2 of the Supplementary material**). Specifically, in the PRO-ACT (vs Lisbon) dataset, the average ALSFRSsUL subscore is 5.81 ± 1.92 (vs 6.36 ± 1.86) during the first appointment and 5.31 ± 2.21 (vs 4.92 ± 2.59) during the third appointment.

Despite the differences, which were somewhat expected given the different nature of the two cohorts, the trajectories revealed in the PRO-ACT dataset confirmed the robustness of our method.

3. Prognostic utility of the assigned clusters – The work can benefit from a survival analysis that uses the assigned clusters.

Answer 1.3: Thank you for your insightful suggestion. We agree that incorporating a survival analysis using the assigned clusters significantly enhances the prognostic utility of our work and provides valuable insights into the potential implications of our clustering approach in predicting patient outcomes.

Addressing the recommendation, we conducted a thorough survival analysis using the assigned clusters. We aimed to elucidate the prognostic significance of the identified trajectories by assessing the time-to-death outcomes within each cluster. This additional analysis was intended to contribute to a more comprehensive understanding of the clinical implications of our stratification model.

The results of this analysis were documented in the subsection “**Survival Analysis in the Lisbon ALS Clinic Dataset**”. We provided a plot of the survival probability over time, together with statistical analysis for each group considering both the proposed ClusTric method and the MoGP framework (in the latter case considering only the 4 most dominant clusters amongst the 27 found by the approach). We further present the mean survival time (in months) and the survival rate at 2 years for each group and compare them with the overall population (**Fig. 4**). In terms of survival curves, we observe that in MoGP only one cluster is statistically different, representing a slower time to death, whereas the other groups have similar average survival times and rates. In contrast, for ClusTric, we observe significant differences between fast (FP), moderate (MP) and slow progressors (SP), with only the two moderate progressors (MPb and MPs) showing non-significant differences, which actually confirms our initial analysis and group namings.

4. *Comparison with MoGP – I feel the comparison between the two methods can be extended to include their prognostic utility. Which of the two methods give better results in predicting future decline?*

Answer 1.4: We appreciate and thank your concern about the survival analysis for the groups identified by the MoGP method. We performed a similar analysis as mentioned in **answer 1.3** for the four most dominant groups identified by MoGP (**Fig 4**), which revealed a group with a higher mean survival time and three groups with similar mean survival times. The ClusTric survival analysis presents a more distinct survival across groups compared to the MoGP corroborating the findings described in Section “**Comparison with MoGP**”, i.e., MoGP does not allow the coherent characterization of the patients within each cluster, as seen by trajectories that strongly overlap over time.

Minor –

I feel the method section can be improved to present a clear mathematical framework of the ClusTric algorithm. This helps the reader in getting a clear picture of the method.

Answer 1.5: Thank you for your suggestion. We made efforts to improve the method section to make it clear to the reader.

Reviewer #2:

The authors modify a previously established tri-clustering algorithm as their base framework for stratifying ALS patient progression based patient-record-time. They apply the unsupervised framework to a single cohort from the Lisbon ALS clinic and describe the resultant characteristics of the four primary clusters and transitions between clusters. They then compare their method to a recently published Gaussian mixture model for ALS progression and stratification.

The work is interesting and significant. It does correctly characterize the non-linearities in ALS that have long been known even though many works often have forced linear assumptions. While the work is interesting and significant, it is not entirely original from a machine learning standpoint given is primarily an application of an existing method. It is original from a domain standpoint given prior work has not used the method in ALS.

The authors explained well the clustering method. The authors correctly state there is not explicit ground truth validation available given this is an unsupervised method.

Answer 2.0: *We are grateful for the favorable remarks made by the reviewer regarding various facets of our proposed approach and the manuscript's quality. Subsequently, we will respond to the highlighted concerns.*

However, there is more the authors could do to assess their results. The biggest drawback to the present manuscript's results is that there is only one cohort assessed. The authors really need to try this method on multiple cohorts and compare them. Comparing the features of the cohort and then comparing the resulting clustering results would provide much more clarity and trust into how well this algorithm works. Using only one cohort does not allow the algorithm to be fairly assessed. There are other open access data available that the authors could use to provide critical baseline assessments. In a journal of this impact, I would consider adding at least one and preferably 2 additional cohorts as a necessity to evaluating the work.

Answer 2.1: Thank you for your insightful observation, which aligns closely with a similar comment from Reviewer 1 (refer also to **answer 1.2**). In that direction, we strived to explore publicly available datasets that closely resemble the characteristics of the Lisbon ALS clinic dataset utilized in our research. Hence, we conducted a parallel analysis using the PRO-ACT database, a publicly accessible ALS cohort. However, it is essential to acknowledge that the statistics of the PRO-ACT dataset significantly differ from those of the Lisbon dataset (see **Table 1** for the characterization of the population in each cohort), as the former originates from ALS clinical trials, where patient selection involves meticulous processes and constraints for trial eligibility. Trial population is characterized by young-onset, better respiratory function and predominant spinal-onset and more men. In addition, patients enrolled in clinical trials have a significantly better prognosis than population-based patients. This introduces inherent biases, particularly when understanding the patient trajectories and evolution in a clinical environment.

Despite notable differences between the two cohorts, four groups still emerged from the PRO-ACT analysis, mirroring those identified in the Lisbon cohort. However, it is crucial to note that the cluster MPs exhibits a slower progression in the Upper Limbs subscore (ALSFRSsUL) and a faster progression in the remaining motor subscores (ALSFRSsT and ALSFRSsLL), see **Table 2 of the Supplementary material** for details. This discrepancy arises because the ALSFRSsUL subscore measurements in the PRO-ACT dataset exhibit much smaller variations over the considered timeframe compared to those observed in the Lisbon cohort. Specifically, in the PRO-ACT cohort (versus the Lisbon cohort), the average ALSFRSsUL subscore is 5.81 ± 1.92 (versus 6.36 ± 1.86) during the first appointment and 5.31 ± 2.21 (versus 4.92 ± 2.59) during the third appointment.

Despite these anticipated differences due to the distinct nature of the two cohorts, the trajectories observed in the PRO-ACT dataset still confirm our initial conclusions, showing the robustness of our method.

The lack of inclusion of static features is another concern that will need fixed in order to maximize the impact of the present work. It appears the authors did perform a post-hoc analysis of static features after clustering. However, it would be better if there was a more

explicit framework for inclusion of static variables. It is possible the authors did have such a framework but, if so, it was not clear based on the currency manuscript.

Answer 2.2: We appreciate your insightful comment. In response to your concern, we performed experiments to include static features in the identification of the groups of patients. Our strategy involved utilizing biclustering to extract patterns from the static features and merging these patterns with the triclustering patterns (representing the temporal patterns). This method did prove to be promising in a prognostic prediction (**classification**) in ALS:

Soares, Diogo F., et al. "Learning prognostic models using a mixture of biclustering and triclustering: Predicting the need for non-Invasive ventilation in Amyotrophic Lateral Sclerosis." *Journal of Biomedical Informatics* 134 (2022): 104172.

However, for the current work (**stratification**), integrating bicluster patterns from static features resulted in a data space increase from 93 to 118 features. The agglomerative clustering algorithm employed in our study uses Euclidean distance as the affinity metric. With the augmented dimensionality, the contrast of the Euclidean distances between different data points decreases, resulting in a negative impact on the clustering results, as also concluded in:

Aggarwal, Charu C., Alexander Hinneburg, and Daniel A. Keim. "On the surprising behavior of distance metrics in high dimensional space." *Database Theory—ICDT 2001: 8th International Conference London, UK, January 4–6, 2001 Proceedings* 8. Springer Berlin Heidelberg, 2001.

In particular, the following figure shows the trajectories of the clusters (for ALSFRS-R, ALSFRSb, ALSFRS_r, and ALSFRS_sUL) resulting from the integration of static features into the proposed approach.

As can be observed, such an approach reveals less coherent progression groups, with static patterns degrading the performance of the clustering algorithm. Hence, we decided to focus the stratification framework on temporal patterns, including the possibility of a posterior analysis regarding the static characteristics of patients in each group. Nonetheless, we have added a paragraph at the end of the Methods section stating that the inclusion of static data through biclustering may be a direction for further research.

While the stratification here is based on progression, it would improve the clinical utility of the work if survival were more overtly incorporated. The authors should provide a censored graph of survival probability over time. Then, they should illustrate survival probability for each of their clusters graphically and then do the follow-up statistical analysis.

Answer 2.3: Thank you for your valuable comment. This comment follows the same concern as addressed in **answer 1.3**. Moreover, we incorporated your suggestions in our survival analysis. We used the time-to-death as the predictive outcome. The analysis is documented in Section “**Survival Analysis in the Lisbon ALS Clinic Dataset**”, and more concretely in **Fig 4**.

I think it is correct for the authors to compare to the recent Gaussian mixture model. However, the authors over sell their results in saying their clustering model is "better" or "faster" without actually providing explicit evidence of these claims. The language of these claims needs toned to be more appropriate.

Answer 2.4: Thank you for your comment. We appreciate your feedback and agree that the language used to describe the performance of our clustering model should be more measured and supported by explicit evidence. We carefully reviewed and modified the language in our manuscript to ensure that our claims accurately reflect the comparative performance of our stratification method.

Reviewer #3:

This manuscript addresses an important area of need in the field of ALS- developing tools to better predict disease progression and stratify patients for clinical trials. The method of combining triclustering and hierarchical clustering to better account for non-linear progression trajectories is innovative and well-described in this manuscript, but will be further assessed by statistical review.

Answer 3.0: *We would like to thank the reviewer for enhancing the potential of our work. Below, we respond to the concerns raised.*

This review will focus on the ALS clinician-scientist perspective. There are a few major concerns with this manuscript in its current form.

1) The authors inappropriately interchange ALS staging systems (static measurements of current disease stage) with ALS projection trajectories (rates of functional decline over time). Staging systems such as MITOS give no information about rate or overall trajectory of progression- the expectation is that each individual person with ALS will progress sequentially through each of the MITOS stages over the course of their disease, but at different rates. In other words, both fast progressors and slow progressors will move through the same MITOS stages in the same order, but the fast progressors move through each stage at a faster rate. It could make sense to compare time spent in various MITOS stages with a progression trajectory, but the current analysis that compares a static MITOS stage with a progression trajectory is not informative.

Answer 3.1: Thank you for this valuable comment regarding the comparison with MITOS stages (a point also similarly raised by reviewer 1 and addressed in **answer 1.1**). Following the reviewers' comments, we removed the study that compared the groups with MITOS stages, given that MITOS gives "no information about rate or overall trajectory of progression".

Moreover, following your observation, we complemented the analysis, including the trajectories of MITOS stages for each disease progression group (**Fig. 1C** and **Fig. 2C**). In fact, we verified that slow progressors only move from stage 1 to stage 2 after 4

appointments while moderate progressors move faster (MPb and MPs started in stage 2, with MPb moving to the next stage within one appointment and MPs within two appointments) and fast progressors always stayed in stage 4 over the considered time-window.

2) *The "Related Work" section appropriately points out the flaws with linear ALS models, but selectively leaves out other more sophisticated modeling approaches that have been more successful. In particular, this section should cite and discuss the work by Berry et. al. "Identifying patterns in amyotrophic lateral sclerosis progression from sparse longitudinal data" and the work of Westeneng et. al. "Prognosis for patients with amyotrophic lateral sclerosis: development and validation of a personalised prediction model."*

Answer 3.2: Thank you for your comment about the cited related work. We analyzed and incorporated a reference to the work of Westeneng et al. that we found really relevant, and we thank you again for pointing it out. Given the journal's publication guidelines, the whole paper was reformatted, and the "Related Work" section was summarized and incorporated in the Introduction section due to space limits in the manuscript.

Regarding the citation to Berry et. al., we believe you are actually referring to the work of Rammamorthy et al., which was already cited in the original submission, and that we already use for comparison purposes (MoGP, see **Figs. 3 and 4** of the revised manuscript). Nonetheless, we also considered a work of Berry et al. as a relevant reference on ALS stratification:

- [6] Berry, J.D., Taylor, A.A., Beaulieu, D., Meng, L., Bian, A., Andrews, J., Keymer, M., Ennist, D.L., Ravina, B.: Improved stratification of als clinical trials using predicted survival. *Annals of clinical and translational neurology* 5(4), 474–485 (2018)

3) *The authors do cite the work of Ramamoorthy and colleagues and compare performance of both models. However, they compare the performance of their 4 proposed trajectories with 27 different trajectories from the Ramamoorthy paper. The Ramamoorthy paper also goes on to identify 4 dominant clinical progression patterns that are not acknowledged in this manuscript, when comparison of both 4-trajectory models would be a more informative comparison.*

Answer 3.3: We appreciate your observation. We agree that a more focused and informative comparison between the proposed trajectories in our study and the 4 dominant clinical progression patterns identified by Ramamoorthy et al. would enhance the clarity and relevance of our findings.

We acknowledge that our initial comparison involved a broad spectrum of 27 trajectories from the Ramamoorthy paper, which might have diluted the specificity of the analysis. In light of your suggestion, we revised our experiment to focus on the direct comparison between our 4 proposed trajectories and the 4 dominant clinical progression patterns identified by Ramamoorthy et al (see **Figs. 3 and 4**). This adjustment allows for a more meaningful and targeted assessment of the performance of our proposed method against this state-of-the-art.

REVIEWER COMMENTS

Reviewer #1 (Remarks to the Author):

I thank the authors for including the suggestions in the last round of reviews. The manuscript looks improved with more supporting evidence. However, I have some concerns regarding the new analysis presented and its limitations.

1. Experiments with PRO-ACT (validation dataset)

a. It was not clear to me if the Clustric algorithm was run again on the PRO-ACT data to get the results presented, or the 3D virtual patterns learned from the Lisbon cohort were used. Using the 3D virtual patterns from the Lisbon dataset shall be preferable since it demonstrates that the representation of disease progression is generalizable to new datasets and hence clinically more meaningful.

b. The progression patterns in the two datasets show differences, especially for the MPs cluster. MPs progression on ALSFRSsUL is very different across the two datasets (Lisbon and PR-ACT). MPb on MITOS is faster than FP. Further, figure 3F and its corresponding sub-figure in 2C (ALSFRS-R for PRO-ACT) do not seem coherent (MPs and MPb seem to progress differently in the two figures for the same dataset.)

c. Minor : The cluster selection criterion are different for the two datasets.

While Clustric shows ability to stratify risk in PRO-ACT, the analysis can be improved to address the above questions.

2. Comparisons with MoGP

My major concern is around how the 4 clusters have been chosen from the MoGP method. The 4 most dominant clusters in MoGP only include 30% of the overall data. Hence they will not have the same ability to capture the overall variance in data (which Clustric can). This could lead to a biased comparison and maybe a reason behind much superior results from Clustric in figure 4. This is not to undermine the good performance of Clustric on survival analysis. However, its advantage over MoGP are not clearly established by this analysis. Similarly, results on table 4 may change if the 4 MoGP clusters were more representative of the entire Lisbon cohort. One way to test this would be

to use a similar agglomerative clustering over all 27 MoGP clusters so the two methods look at identical data.

3. Transitions across clusters

Transitions across clusters adversely effect the prognostic utility of the model. Given that a significant fraction of the population shows transitions across clusters (especially from SP to MPb and MPs), the work should highlight the limitations of the present 4 clusters in capturing the variation of progression patterns. More clusters may be needed to capture such composite progression patterns where the progression trajectories of subjects switches across clusters during the overall progression.

Reviewer #2 (Remarks to the Author):

The authors have greatly improved the manuscript from its original submission. The major concerns were addressed. Key improvements include:

1. Addition of a much needed second cohort
2. Addition of survival analysis of the clusters
3. Improved clarity of over all paper - including better method clarity, overall structure, and improve comparisons to related work and baselines.

There is only one point that remains a minor concern. The reviewers do address the issue of not including the static features in their clustering and say that this is an area for future work. Another sentence or two is needed to better explain WHY static features mathematically result in a degradation of features. Such an explanation should illustrate that the inclusion of static features resulting in degraded performance is a methodological limitation. Otherwise, with the current wording, it could incorrectly be interpreted by domain experts that static features actually and literally impede the viewed clarity of disease progression.

Reviewer #3 (Remarks to the Author):

Overall, this revised manuscript is much improved compared to the original version and does an excellent job of incorporating reviewer feedback. The external validation of the model using the PRO-ACT dataset and the survival analyses are particular strengths in this revision.

Additional revision is recommended for the discussion section. The last paragraph of the discussion should be proofread and edited for clarity. While it is important to note the differences between the Lisbon and PRO-ACT cohorts, this should not be the final take-home point of the discussion section.

The discussion section also needs a limitations section. Two notable limitations are 1) This approach is likely not feasible to stratify clinical trial participants at the time of enrollment since it requires 3 assessments over a 6-month period. Most clinical trials seek to enroll participants shortly after diagnosis. 2) The fact that 1/3 of patients change trajectories after a 6-month period is substantial and could very much limit the use of this model in a clinical trial or clinical setting.

Response to Reviewers

General Comments to Reviewers:

Thank you for all your attention, detailed feedback, and for providing us the opportunity to resubmit an improved version of our manuscript, **NCOMMS-23-40504A**. This document details how the raised concerns were incorporated into the revised version of the manuscript.

While revising the manuscript we identified an error in our preprocessing of PRO-ACT dataset, more precisely, in the computation of the ALSFRSsUL subscore. In PRO-ACT there are two features associated with the ALSFRS question 5 (5a. Cutting without Gastrostomy and 5b. Cutting with Gastrostomy). We wrongly assumed that these questions were mutually exclusive (as it happens in the Lisbon ALS Clinic dataset), that is, there were no subjects with non-zero values in the two questions simultaneously. However, when revising the experiments, we noticed that 23 snapshots out of the 37031 total snapshots in PRO-ACT had non-zero values in the two features at the same time. Considering only the first 3 consecutive assessments the number of affected snapshots was 14 out of 12132 (corresponding to 10 patients from the total 3880 patients used in our experiments). To fix this problem, we re-run all the experiments performed with PRO-ACT. This led to minor changes in the results and did not impact the main conclusions. The text changes resulting from fixing this issue are highlighted in **red**. Fig. 2 and Fig.3F were updated due to this correction.

The manuscript was carefully revised to address all the received comments and proof-checked.

The changes to the revision of the previous manuscript are highlighted in **blue**.

Reviewer #1

I thank the authors for including the suggestions in the last round of reviews. The manuscript looks improved with more supporting evidence. However, I have some concerns regarding the new analysis presented and its limitations.

Answer 1.0: We thank the reviewer for the comprehensive review and the positive comments regarding the improvements made based on the previous round of reviews. Below, we respond to the concerns raised.

1. Experiments with PRO-ACT (validation dataset)

a. It was not clear to me if the ClusTric algorithm was run again on the PRO-ACT data to get the results presented, or the 3D virtual patterns learned from the Lisbon cohort were used. Using the 3D virtual patterns from the Lisbon dataset shall be preferable since it demonstrates that the representation of disease progression is generalizable to new datasets and hence clinically more meaningful.

Answer 1.1a: *Thank you for the valuable insight regarding the experiment with PRO-ACT.*

In the manuscript, the ClusTric algorithm was completely re-run on the PRO-ACT dataset, thus using PRO-ACT extracted patterns (as clarified at the beginning of section “ClusTric in PRO-ACT Dataset”). ClusTric is proposed as a temporal clustering algorithm and used for patient stratification, and not for a transfer learning setting. Thus, we focus on validating its strength in different cohorts, in this case, one ALS dataset was collected in general clinical settings, and the other was collected for clinical trials. Furthermore, the specific patient selection criteria used in the clinical trials leading to PRO-ACT dataset reduces patient heterogeneity observed in clinical practice as evidenced by the clustering results.

b. The progression patterns in the two datasets show differences, especially for the MPs cluster. MPs progression on ALSFRSsUL is very different across the two datasets (Lisbon and PRO-ACT). MPb on MITOS is faster than FP.

Answer 1.1b: *The reviewer is right about the differences between disease progression trajectories for the MPs groups in the two cohorts. This is explained by the patient inclusion criteria used in the clinical trials leading to PRO-ACT dataset. As stated in the Discussion Section: “Our results underline the differences between clinical trial-based vs hospital-based ALS patients. Clinical trial population is characterized by young-onset, preserved respiratory function, more men, and predominant spinal-onset phenotype. In addition, patients enrolled in clinical trials have in general a significantly better prognosis than a hospital-based population [29]. In particular, patients enrolled in PRO-ACT have higher body mass index, and more frequent upper limb weakness with a slower disease spreading (a phenotype more common in young men [30]). Patients with bulbar-onset and without other regions affected are not candidates for clinical trials, some of these patients progress slowly to other regions (slower spreading). This supports a higher ALSFRS-R observed in hospital-based patients when compared to PRO-ACT bulbar-onset patients, representing two different populations of bulbar-onset patients. Clearly, the hospital-based Lisbon population was more heterogeneous, including older-patients with axial-onset, not eligible for clinical trials due to the poor respiratory tests associated with this phenotype.”*

These differences between the two datasets result in different disease progression patterns. Despite these differences, clusters with similar characteristics emerged.

Further, figure 3F and its corresponding sub-figure in 2C (ALSFRS-R for PRO-ACT) do not seem coherent (MPs and MPb seem to progress differently in the two figures for the same dataset.)

Answer 1.1b: Figures 3F and 2C (ALSFRS-R for PRO-ACT) differ due to the preprocessing steps required by MoGP, which were also performed before applying ClusTric to the PRO-ACT dataset to allow a fair comparison between the two algorithms. These preprocessing steps eliminate patients. Consequently, the curves shown in Figure 3F, using the reduced set of patients, are slightly different from the ones in Figure 2C, where the whole dataset was used. Additionally, we applied a smoothing technique to the trajectories of Figure 3F to compare them with the smoothed trajectories generated by MoGP. To make it clear, we have modified the caption of Figure 3 to mention the preprocessing carried out on both cohorts.

Furthermore, we note that Figure 3F shows 12 follow-up appointments while Figure 2C only shows the first 5 appointments. In this context, for comparison purposes, we truncated the curve in Figure 3F to the first 5 appointments (see first figure (A)) and applied the smoothing technique to Figure 2C (second figure (B)). We can include these figures in Figure 3 if the review considers them useful.

c. Minor : The cluster selection criterion are different for the two datasets.

Answer 1.1c: *The reviewer is right. We revised the manuscript to consider the same cluster selection criteria in both cohorts, namely Figure 3 and the first paragraphs in Sections “ClusTric in Lisbon ALS Clinic Dataset” and “ClusTric in PRO-ACT Dataset”.*

While Clustric shows ability to stratify risk in PRO-ACT, the analysis can be improved to address the above questions.

2. Comparisons with MoGP

My major concern is around how the 4 clusters have been chosen from the MoGP method. The 4 most dominant clusters in MoGP only include 30% of the overall data. Hence they will not have the same ability to capture the overall variance in data (which Clustric can). This could lead to a biased comparison and maybe a reason behind much superior results from Clustric in figure 4. This is not to undermine the good performance of Clustric on survival analysis. However, its advantage over MoGP are not clearly established by this analysis.

Similarly, results on table 4 may change if the 4 MoGP clusters were more representative of the entire Lisbon cohort.

One way to test this would be to use a similar agglomerative clustering over all 27 MoGP clusters so the two methods look at identical data.

Answer 1.2: *Regarding your major concern about how the 4 clusters were chosen from the MoGP method, we would like to clarify that these clusters were selected based on their size, mimicking the procedure in the MoGP paper. They thus correspond to the four largest clusters out of the 27 clusters identified by the MoGP method. We further note that in MoGP paper the authors also present the survival curves for the five largest clusters and these 5 clusters only include around 15% of the overall data. We agree that with such a low representation of the population, those clusters do not have the same ability to capture the overall variance in data, but this is a limitation of MoGP, needing 27 clusters to represent the entire population (some trajectories have only one patient and 4 trajectories have between 1 and 10 patients) while ClusTric needs only four clusters.*

We changed Table 4 to include, in the case of MoGP, the metrics for the 4 most dominant clusters besides those that were already presented for all clusters in both datasets.

Regarding the application of an agglomerative clustering algorithm over the MoGP clusters, we would like to point out that in ClusTric we are not clustering trajectories of patients but clustering a transformed version of the data relying on disease progression patterns discovered with triclustering. Furthermore, the inclusion of an agglomerative clustering step to MoGP corresponds to proposing an improved version of that method, which is out of the scope of this paper.

3. Transitions across clusters

Transitions across clusters adversely effect the prognostic utility of the model. Given that a significant fraction of the population shows transitions across clusters (especially from SP to MPb and MPs), the work should highlight the limitations of the present 4 clusters in capturing the variation of progression patterns. More clusters may be needed to capture such composite progression patterns where the progression trajectories of subjects switches across clusters during the overall progression.

Answer 1.3: *The clustering results are based on 3 consecutive appointments and the progression groups are thus obtained considering that period. In the transition study, we predict what will be the progression group for each patient in appointments 3 to 5 (second period) training a classifier using the clustering results as the class label. Thus, assuming the progression group does not change throughout disease progression. The transition between groups (“classification errors”) reflects two realities: (i) typical classification errors; or (ii) actual changes in the patient's disease progression, corresponding to a “classifier error” that in reality is not an error since the patient actually evolves differently. It is well-known in clinical practice that a small number of ALS patients deviate from the expected disease progression [34]. Their disease progression is frequently non-linear, limiting the accuracy of predictive models, since some of these patients can progress more or less rapidly than previewed [8][35]. We added this discussion to the Discussion section.*

Regarding the increase of the number of clusters we note that given that the disease progression is coherent with those of the group in the time window used for clustering, the group changes could only be captured by the clustering algorithm if more appointments were considered to compute the groups.

We further added the percentages to the table in Figure 5.A, to make it clear that the number of changes per group is small. In the examples below, we show the ALSFRS-R of 4 patients changing their progression group. Figures a) and b) show patients who were initially classified as slow progressors (almost constant score) but in the second period, their progression changed to a moderate progression. An identical behavior is observed in Figures c) and d) where the patients with moderate progression in the first period change to a faster progression in the second period.

a)

b)

c)

d)

[34] Requardt, M. V., Görlich, D., Grehl, T., & Boentert, M. (2021). Clinical determinants of disease progression in amyotrophic lateral sclerosis—a retrospective cohort study. *Journal of Clinical Medicine*, 10(8), 1623.

[8] Ramamoorthy D, Severson K, Ghosh S, Sachs K, Glass JD, Fournier CN, Pooled Resource Open-Access ALS Clinical Trials Consortium Sherman Alexander 5, Herrington TM, Berry JD. Identifying patterns in amyotrophic lateral sclerosis progression from sparse longitudinal data. *Nature Computational Science*. 2022 Sep;2(9):605-16.

[35] Gordon, P. H., Cheng, B., Salachas, F., Pradat, P. F., Bruneteau, G., Corcia, P., ... & Meininger, V. (2010). Progression in ALS is not linear but is curvilinear. *Journal of neurology*, 257, 1713-1717.

Reviewer #2

The authors have greatly improved the manuscript from its original submission. The major concerns were addressed. Key improvements include:

1. Addition of a much needed second cohort
2. Addition of survival analysis of the clusters
3. Improved clarity of over all paper - including better method clarity, overall structure, and improve comparisons to related work and baselines.

Answer 2.0: We would like to thank the reviewer for the positive feedback and are glad to hear that the major concerns have been addressed.

There is only one point that remains a minor concern. The reviewers do address the issue of not including the static features in their clustering and say that this is an area for future work. Another sentence or two is needed to better explain WHY static features mathematically result in a degradation of features. Such an explanation should illustrate that the inclusion of static features resulting in degraded performance is a methodological limitation. Otherwise, with the current wording, it could incorrectly be interpreted by domain experts that static features actually and literally impede the viewed clarity of disease progression.

Answer 2.1: We thank the reviewer for insisting on this point. This leads us to doubt our previous conclusions regarding static features and to perform further experiments. During this process, we identified an error in the dataset in the static feature "Diagnostic Delay" which we now corrected and performed the following set of experiments now presented in section 1 of the Supplementary Material, called "The role of static features in ALS stratification with ClusTric":

- 1) *Stratification using static features and three appointments*
- 2) *Stratification using the first appointment*

Although static features can be used together with the three appointments leading to the progression groups in Fig. S1, their inclusion results in less coherent groups since the higher patients' heterogeneity reflected in these features is not translated into the same heterogeneity when analysing disease progression using the functional scores. For this reason, we decided to maintain the results without static features in the main paper, but include all the results with static features together with a discussion on their impact in the Supplementary Material. We also performed experiments using only the first appointment with and without static features, which confirmed that ALSFRS features alone better describe the different disease progression groups. We added a paragraph to the Discussion section about the impact of static features in ALS stratification clarifying that these might not be the case in other diseases.

Reviewer #3

Overall, this revised manuscript is much improved compared to the original version and does an excellent job of incorporating reviewer feedback. The external validation of the model using the PRO-ACT dataset and the survival analyses are particular strengths in this revision.

Answer 3.0: Thank you for your positive feedback on the revised manuscript. We appreciate your acknowledgment of the improvements made by incorporating the previous feedback. Subsequently, we will respond to the highlighted concerns.

Additional revision is recommended for the discussion section. The last paragraph of the discussion should be proofread and edited for clarity. While it is important to note the differences between the Lisbon and PRO-ACT cohorts, this should not be the final take-home point of the discussion section.

Answer 3.1: We revised all the Discussion section, in particular the last paragraph with the take-home lesson about our proposed method.

The discussion section also needs a limitations section. Two notable limitations are 1) This approach is likely not feasible to stratify clinical trial participants at the time of enrollment since it requires 3 assessments over a 6-month period. Most clinical trials seek to enroll participants shortly after diagnosis. 2) The fact that 1/3 of patients change trajectories after a 6-month period is substantial and could very much limit the use of this model in a clinical trial or clinical setting.

Answer 3.2: We included some paragraphs in the revised version of the Discussion to address the concerns raised regarding the number of assessments required by our approach and the transition study.

Regarding the first concern, we clarified in the Discussion section that ClusTric is flexible in working with any number of appointments. In our case study, we decided to set the number of consecutive appointments to 3 to study disease progression only captured in temporal data dynamics. We acknowledge that the number of assessments before enrollment depends on each specific clinical trial, and only one assessment may be required. In this case, ClusTric can be applied with only one assessment. We clarified this point in the Methods section. Moreover, we performed additional experiments using just the first patients' appointment and presented these results as Supplementary Material. As expected, when only one appointment is used, the obtained groups reflect the differences in the patients' functional state (with specific domains affected) and do not capture disease progression. We clarified this point in the Discussion section.

Regarding concern 2), the clustering results are based on 3 consecutive appointments and the progression groups are thus obtained considering that period. In the transition study, we predict what will be the progression group for each patient in appointments 3 to 5 (second period) training a classifier using the clustering results as the class label. Thus, assuming the progression group does not change throughout disease progression. The transition between

groups (“classification errors”) reflects two realities: (i) typical classification errors; or (ii) actual changes in the patient's disease progression, corresponding to a “classifier error” that in reality is not an error since the patient actually evolves differently. It is well-known in clinical practice that a small number of patients deviate from the expected disease progression. We further added the percentages to the table in Figure 5.A, to make it clear that the number of changes per group is small, and in the particular case of transitioning to a group with slower progression is almost residual (refer to Table of Figure 5.A). We also added a paragraph in the Discussion section. The observed transitions are associated with the heterogeneous nature of ALS which is exemplified by **Figures a), b), c), and d)** of **answer 1.3**, also shown below:

These Figures show patients who initially had a slow disease progression and in the second period changed to moderate progressors (a) and b)) and patients with moderate progressions who exhibited a fast disease progression after 6 months (c) and d)).

REVIEWERS' COMMENTS

Reviewer #1 (Remarks to the Author):

I find the work substantially improved. Changes to figure 5 are a positive addition, that helps to better understand the heterogeneity in disease progression patterns. Comparisons with MoGP show that ClusTric does better at capturing diverse progression patterns. These improvements should be seen in the context of the limitations of MoGP. The authors have used the top 4 MoGP clusters, but that does not ensure the most diverse MoGP clusters being captured. This however, is a limitation of the MoGP method.

Regarding the differences in progression patterns across the datasets, it is plausible that the differences are partly due to the selection criterion for clinical trials in the PRO-ACT dataset. However, it is not clear why that is the only reason which may explain this difference.

An added analysis which could be included, is the survival analysis (kaplan-meier) for the PRO-ACT dataset to highlight the similarity and differences across the two datasets.

Reviewer #2 (Remarks to the Author):

The authors have made substantive additional revisions that mostly address the requested key technical revisions and clarity concerns. I have no further requests at this time.

Reviewer #2 (Remarks on code availability):

Readme file present and comments are adequate.

Reviewer #3 (Remarks to the Author):

This revised manuscript adequately addresses reviewer feedback and is much improved compared to prior versions. From the standpoint of clinical review, no additional changes are required.

Response to Reviewers

Reviewer #1

I find the work substantially improved. Changes to figure 5 are a positive addition, that helps to better understand the heterogeneity in disease progression patterns. Comparisons with MoGP show that ClusTric does better at capturing diverse progression patterns. These improvements should be seen in the context of the limitations of MoGP. The authors have used the top 4 MoGP clusters, but that does not ensure the most diverse MoGP clusters being captured. This however, is a limitation of the MoGP method.

Answer 1.0: Thank you for your positive feedback on our revised manuscript.

Regarding the differences in progression patterns across the datasets, it is plausible that the differences are partly due to the selection criterion for clinical trials in the PRO-ACT dataset. However, it is not clear why that is the only reason which may explain this difference.

Answer 1.1: The selection criteria impose a significant difference in this case. It makes the PRO-ACT patients more homogenous, leading to less distinct patterns and consequently slightly different trajectories. We recall that the clinical trial population is characterized by young-onset, preserved respiratory function, and a predominant spinal-onset phenotype. In general, patients have a significantly better prognosis than those of a hospital-based population [29]. When compared to the Lisbon Clinical Dataset, patients enrolled in PRO-ACT have higher body mass index and more frequent upper limb weakness with a slower disease spreading (a phenotype also more common in young men [30]).

A striking difference between the two datasets that strongly impacts the patterns found by the algorithm as well as the cluster trajectories is that patients with bulbar-onset and without other regions affected are not candidates for clinical trials, although some of these patients progress slowly to other regions (slower spreading).

In summary, the hospital-based Lisbon population is more heterogeneous, including older patients with axial-onset, not eligible for clinical trials due to the poor respiratory tests associated with this phenotype. It leads to differences in patterns and trajectories that are well highlighted in the manuscript.

An added analysis which could be included, is the survival analysis (kaplan-meier) for the PRO-ACT dataset to highlight the similarity and differences across the two datasets.

Answer 1.2: The survival analysis for the PRO-ACT dataset is presented in the figure below.

When compared with the results reported in the manuscript (fig. 4 - ClusTric), it can be observed that, while the SP group remains with the highest survival probability (compared to the other groups), the mean survival time is different: 35 months in PRO-ACT vs 54 months in the Lisbon Dataset. However, we also notice that the SP group includes ~88% of censored records in PRO-ACT, which compares with ~40% of censored records in the Lisbon dataset.

Another difference is observed in the bulbar group, which is highly affected by the inclusion criteria (we recall that patients with bulbar-onset and without other regions affected are not candidates for clinical trials). The mean survival time of the MPb group is ~22 months on PRO-ACT (69% censored) vs ~32 months on the Lisbon dataset (21% censored).

The significant differences in censored records make a fair comparison impossible. So, we decided not to include this figure in the manuscript's supplementary materials.

Reviewer #2

(Remarks to the Author):

The authors have made substantive additional revisions that mostly address the requested key technical revisions and clarity concerns. I have no further requests at this time.

Answer 2.0: Thank you for your positive feedback. We are glad to hear that all the previously mentioned concerns were addressed.

(Remarks on code availability):

Readme file present and comments are adequate.

Answer 2.1: Thank you for your positive feedback regarding the provided code.

Reviewer #3

This revised manuscript adequately addresses reviewer feedback and is much improved compared to prior versions. From the standpoint of clinical review, no additional changes are required.

Answer 3.0: Thank you for your positive feedback on the revised manuscript. We appreciate your acknowledgement of the improvements made by incorporating the previous feedback.